# A novel PLpro inhibitor improves outcomes in a pre-clinical model of long COVID

The COVID-19 pandemic caused by the coronavirus SARS-CoV-2 has highlighted the vulnerability of a globally connected population to zoonotic viruses. The FDA-approved coronavirus antiviral Paxlovid targets the essential SARS-CoV-2 main protease, Mpro. Whilst effective in the acute phase of a COVID infection, Paxlovid cannot be used by all patients, can lead to viral recurrence, and does not protect against post-acute sequelae of COVID-19 (PASC), commonly known as long COVID, an emerging significant health burden that remains poorly understood and untreated. Alternative antivirals that are addressing broader patient needs are urgently required. We here report our drug discovery efforts to target PLpro, a further essential coronaviral protease, for which we report a novel chemical scaffold that targets SARS-CoV-2 PLpro with low nanomolar activity, and which exhibits activity against PLpro of other pathogenic coronaviruses. Our lead compound shows excellent in vivo efficacy in a mouse model of severe acute disease. Importantly, our mouse model recapitulates long-term pathologies matching closely those seen in PASC patients. Our lead compound offers protection against a range of PASC symptoms in this model, prevents lung pathology and reduces brain dysfunction. This provides proof-of-principle that PLpro inhibition may have clinical relevance for PASC prevention and treatment going forward.

The COVID-19 pandemic has to date seen >775 million people infected with Severe Acute Respiratory Syndrome Coronavirus-2 (SARS-CoV-2), leading to 7 million deaths[1]. Coronaviruses (CoVs) were first recognised to circulate in humans in the 1960s[2], and despite deadly outbreaks of SARS-CoV in 2003, Middle East Respiratory Syndrome (MERS) CoV in 2012, and circulation of further CoVs with milder human pathologies (HKU1, OC43, NL63 and 229E), the world was unprepared to counter SARS-CoV-2. Unprecedented scientific efforts and the fastest vaccine development ever achieved[3], have softened the impact of COVID-19 on the global population, yet SARS-CoV-2 remains a major cause of death in humans[4], and a new pandemic of its long-term effects has emerged.

Post-acute sequelae of COVID-19 (PASC) affects approximately one-third of non-hospitalised COVID-19 patients. It is estimated that over 77 million people have experienced continuation or development of new symptoms after the initial SARS-CoV-2 infection[5]. PASC presents a complex and persistent health condition that is primarily driven by a sustained dysregulated host immune response and continues to affect patients with a wide range of physical and psychological consequences[6]. Collectively, PASC not only significantly affects individual patients, but also impacts the healthcare system and societal structures[7]. There are currently no available treatments for patients suffering from PASC[8].

The lack of effective antivirals against CoVs early in the pandemic was problematic and preventable. The first antiviral treatments approved for emergency use for COVID-19 included RNA polymerase inhibitors remdesivir[9] and molnupiravir[10], however, their effectiveness was limited[11]. A more efficacious antiviral, Paxlovid, was approved in late 2021[12] and has since been widely used. Paxlovid targets the essential CoV main protease (Mpro), an enzyme required to cleave viral polypeptides into functional proteins. In Paxlovid, the Mpro inhibitor nirmatrelvir[12] is paired with the pharmacokinetic enhancer ritonavir[13], a potent CYP3A4

✉ e-mail: devine.s@wehi.edu.au; glessene@wehi.edu.au; doerflinger.m@wehi.edu.au; dk@wehi.edu.au

inhibitor required to extend the half-life of nirmatrelvir. However, due to this CYP3A4 inhibition not all patients can safely access Paxlovid, with potential for unwanted drug-drug interactions. Antivirals without this CYP3A liability would expand treatment options to greater patient cohorts. Paxlovid has not shown any efficacy in preventing or reversing PASC[14]. Alternative Mpro inhibitors have been reported[15,16] but so far, none have been approved by the FDA.

In addition to Mpro, coronaviruses encode an essential papain-like protease (PLpro) within non-structural protein 3 (nsp-3). PLpro processes a different set of viral polyproteins, and in addition acts as a deubiquitinase (DUB) and deISGylase[17,18]. The latter functions directly suppress the host immune response[19]. It has been speculated that PLpro inhibition prevents not only viral replication but may also prevent viral interference with inflammation cascades that contribute to observed immune phenotypes. Efficacious PLpro small molecule inhibitors, all based on the GRL0617 (hereafter GRL-) scaffold first developed as SARS-CoV PLpro inhibitors[20], have recently shown in vivo efficacy in SARS-CoV-2 animal models[21,22] while other compounds indicated efficacy across diverse CoVs[23]. One GRL-like PLpro inhibitor (HL-21) has recently advanced to clinical studies[24,25]. In this study, we describe a novel chemical scaffold that targets the essential coronavirus protease PLpro, with low nanomolar activity against SARS-CoV-2 PLpro and other pathogenic coronaviruses. Our lead compound shows excellent efficacy in a mouse model of severe acute disease and prevents key long-term pathologies associated with PASC, including lung and brain dysfunction. Our work provides proof-of-principle that PLpro inhibition may be clinically relevant for PASC prevention and treatment.

## Results
### The WEHI-P series represent a novel scaffold for PLpro inhibitors

To identify novel inhibitors of SARS-CoV-2 PLpro, we utilised our Ubiquitin (Ub) Rhodamine110 (UbRh)-based high throughput screening platform[18,26,27] to screen a diverse library of >400,000 small molecules (Fig. 1a, Supplementary Fig. 1a–c). The screening campaign arrived at 16 hit compounds that were validated in an orthogonal surface plasmon resonance (SPR) assay, warranting further investigation (Fig. 1a). Binding of GRL-scaffold PLpro inhibitors relies on a so-called 'blocking loop-2' (BL2) sequence, that encloses inhibitors in a binding groove near the catalytic centre[18,26,28,29] (see below). To identify the most interesting compounds from our screen, we counter-screened compounds against SARS-CoV, MERS-CoV and a SARS-CoV-2 PLpro BL2 variant (termed SARS-CoV-2 PLpro$^{BL}$) in which the SARS-CoV-2 BL2 loop (sequence G-NYQC-G) was replaced with the MERS-CoV BL2 loop (sequence G-IETAV-G) (Supplementary Fig. 2a, **Methods**. **WEHI-P1** (Fig. 1b) inhibited SARS-CoV-2 PLpro with an IC$_{50}$ of 2.6 μM, and interestingly, also inhibited SARS-CoV-2 PLpro$^{BL}$ with an IC$_{50}$ of 3.4 μM, suggesting a distinct binding mode from known scaffolds (Supplementary Fig. 2a). **WEHI-P1** showed no cross-reactivity in a panel of human DUBs (Supplementary Fig. 2a). To support the subsequent hit-to-lead programme, **WEHI-P** series compounds were assessed in a panel of biochemical, cellular and antiviral screens, including UbRh inhibition (IC$_{50}$); direct compound binding (SPR, $K_D$); inhibition of cleavage of an in-cell FRET-based biosensor (cellular EC$_{50}$[30],); and antiviral efficacy in inhibiting viral replication (plaque assay, antiviral EC$_{50}$) (Fig. 1b, Supplementary Fig. 2). Medicinal chemistry efforts improved **WEHI-P** series compounds (Fig. 1b), and biopharmaceutical properties were monitored throughout the programme.

The replacement of the central ketone group with an oxime led to **WEHI-P2**, with a submicromolar IC$_{50}$ and with micromolar cellular activity in a FRET assay and was further elaborated to introduce a cyclohexanol group to generate **WEHI-P3**. When tested at 50 μM concentration, **WEHI-P3** did not show any inhibitory activity against 47 human DUBs from diverse families (Supplementary Fig. 2h).

Enantiomer separation of **WEHI-P3** gave **WEHI-P4** which exhibited an IC$_{50}$ of 19 nM, a $K_D$ of 18 nM, and a cellular EC$_{50}$ of 310 nM (Fig. 1b, Supplementary Fig. 2). Next, we evaluated the median tissue culture infection dose (TCID50) of SARS-CoV-2 on the human lung epithelial cell line Calu-3[27,31,32] in the presence of nirmatrelvir, the active Mpro inhibitor ingredient of Paxlovid, and **WEHI-P4** (Supplementary Fig. 2d). While a concentration of 2 μM nirmatrelvir reduced the viral titre by 2-log, the same concentration of **WEHI-P4** reduced the viral titre almost to the limit of detection. Importantly, while 0.5 μM nirmatrelvir had no antiviral effect, 0.5 μM **WEHI-P4** reduced viral titres by >2-log (Supplementary Fig. 2d). In order to quantify antiviral EC$_{50}$, we performed Vero plaque live virus infection assays. Importantly, **WEHI-P4** displayed an antiviral EC$_{50}$ of 410 nM in a plaque assay (Fig. 1b, Supplementary Fig. 2e, f). Nirmatrelvir in this assay, requires the addition of a P-glycoprotein (Pgp) inhibitor[33] to achieve a comparable efficacy with an antiviral EC$_{50}$ of 530 nM (>2 μM without Pgp inhibitor) (Supplementary Fig. 2e, f). Of note, our **WEHI-P4** compound does not require auxiliary Pgp inhibition for its antiviral efficacy, however we saw a slight improvement (EC$_{50}$ = 306 nM) when co-treated (Supplementary Fig. 2e, f).

### WEHI-P series compounds exploit a distinct binding site on PLpro

Most medicinal chemistry campaigns targeting PLpro have focused on improving GRL-scaffold inhibitors, which bind PLpro in a binding groove near the active site, forming extensive interactions with BL2[34]. A 1.98 Å co-crystal structure of original hit **WEHI-P1** bound to PLpro (Fig. 1c, Supplementary Fig. 3a, b, Supplementary Table 1) revealed how the compound occupied the GRL-binding site only partially, and instead used a pronounced hydrophobic pocket absent in any other published PLpro inhibitor structures[21,26] including that of PLpro bound to GRL0617[34] (Fig. 1c, d). This new binding site arises from rotation of the side chain of Met208, allowing the methoxynaphthalene of **WEHI-P1** to sit atop Pro247 (Fig. 1c, d). This dynamic behaviour of Met208 has not been reported in the >30 compound crystal structures for this enzyme and opens new opportunities for drug design. The ketone group of **WEHI-P1**, points towards the BL2-groove, which has been exploited in recent more potent GRL-analogues[21,35]. The nitrogen on the piperidine ring forms an essential hydrogen bond with Asp164, a contact also known to be crucial for GRL-like compounds. Consistent with our biochemical data, the BL2 loop of PLpro did not contact the ligand in our **WEHI-P1** complex structure. This enables the BL2 loop to open in this crystal form (space group $P2\,2_1\,2_1$) and form tight interactions with a second PLpro molecule in the asymmetric unit (ASU). In this space group, the inhibitor is bound between the two molecules (Supplementary Fig. 3b, c). **WEHI-P**-series compounds however do not induce PLpro dimerisation in size exclusion chromatography coupled to multi angle light scattering (SEC-MALS) studies (Supplementary Fig. 3d).

Moreover, co-crystals with **WEHI-P2** in the same space group failed to explain the significantly improved IC$_{50}$ achieved through oxime incorporation; the oxime group, installed to engage the BL2 groove, did not form protein contacts and was poorly defined by electron density. We hence determined a 2.8 Å co-crystal structure of the more potent **WEHI-P4** in another space group ($P4_3\,3\,2$) (Fig. 1c, d, Supplementary Fig. 3a, b, e, Supplementary Table 1), with one molecule in the ASU, no significant crystal contacts near the compound binding site, and with a closed BL2-loop that contributes to compound binding. In this structure, **WEHI-P4** displayed two additional interactions: the side chain of BL2-loop Tyr268 stacks against the oxime group of the compound, and Tyr268 forms a backbone hydrogen bond with the hydroxy group of the cyclohexanol moiety (Fig. 1c, Supplementary Fig. 3a, e). Our structures collectively reveal how **WEHI-P** series compounds achieve their high potency through exploiting an unexpected PLpro binding site, expanding PLpro drug discovery to a new chemical scaffold.

## Towards broad-spectrum PLpro inhibitors

To generate antivirals with broad utility and expanded future use, we considered two layers. Firstly, we note that PLpro has remained remarkably evolutionarily stable across SARS-CoV-2 variants. Mutations in PLpro that have arisen in SARS-CoV-2 variants-of-concern are all remote from our compound binding site and not expected to impact inhibitor binding (Supplementary Fig. 4a).

Secondly, we were keen to understand whether **WEHI-P** series compounds could target PLpro enzymes from other CoVs, as it would be beneficial to develop broadly active CoV antivirals to support efforts towards pandemic preparedness. Genetic diversity among CoVs is significant, but DUB and deISGylase activity in PLpro enzymes is a conserved feature across CoVs[36]. Indeed, αCoVs harbour two PLpro domains in nsp-3, of which the PL2pro domain shows highest similarity to SARS-CoV-2 PLpro, and were recently confirmed to be DUBs[36]. We performed sequence comparisons for the seven CoVs that are known to infect humans, namely βCoVs

SARS-CoV-2, SARS-CoV, MERS, HKU1 and OC43, as well as αCoVs NL63 and 229E (Supplementary Figs. 4b, 5), and compiled their structures from the Protein Data Bank (PDB) and AlphaFold2[37] (Fig. 2a), expressed and purified the PLpro enzymes (Supplementary Fig. 4c) and established UbRh cleavage assays for each PLpro (Supplementary Fig. 4d, e) (see **Methods**. All studied PLpro domains comprise DUB activity.

We next tested a small yet diverse subset of our **WEHI-P** series compounds against all seven enzymes, to understand whether any compound displayed pan-PLpro activity. **WEHI-P4** inhibited SARS-CoV-2 and SARS-CoV, and surprisingly also showed weak activity (IC$_{50}$ 52 μM) against PL2pro of the αCoV NL63, which has only 20% sequence identity to SARS-CoV-2 (Fig. 2b, Supplementary Fig. 4d, e). Replacing the cyclohexanol with a 3-substituted pyrazole in **WEHI-P70** strengthened pan activity, inhibiting four of the seven PLpro enzymes with nanomolar activity against SARS-CoV-2 and SARS-CoV (IC$_{50}$ 98 nM and 94 nM, respectively), submicromolar activity against

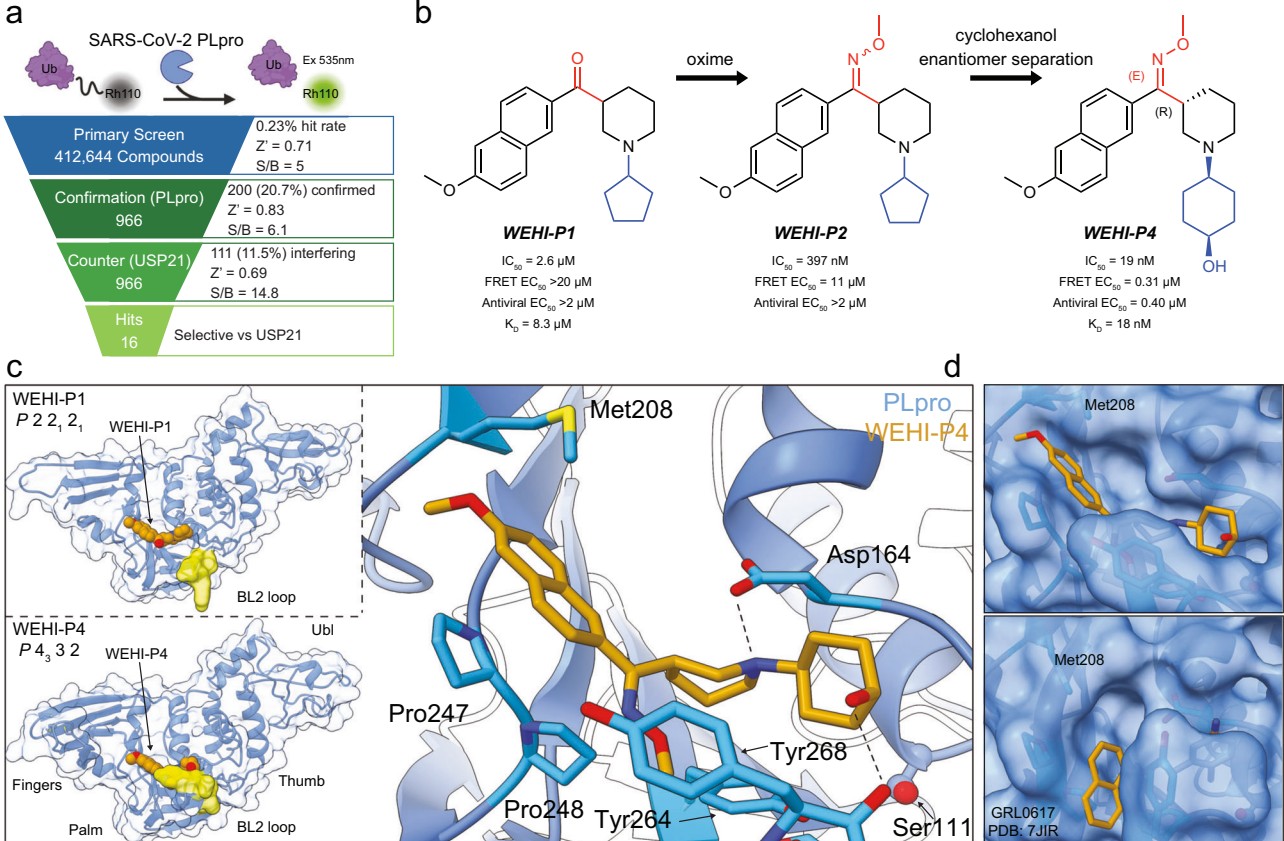

**Fig. 1 | The *WEHI-P* series is a novel scaffold for PLpro inhibitors. a** High Throughput Screening cascade for the identification of PLpro inhibitors. Schematic top, Ub-Rhodamine110 (UbRh) assay was used for assessing the biochemical inhibition (IC$_{50}$) of PLpro. Rhodamine110 is cleaved off the ubiquitin moiety by PLpro which releases a fluorescent signal that accumulates proportional to activity and can be measured at 535 nm. Below, a diverse library of 412,644 compounds was screened in a single concentration (29.16 μM) with one replicate. 966 compounds (hit rate of 0.23%) were identified from the primary screen. These compounds were then assessed in a 10-point titration study in the PLpro assay (confirmation) and USP21 assay (counter) in duplicate. 20.7% of primary hits, or 200 compounds, had confirmed activity in the PLpro assay. 11.5% of hits from the primary screen, or 111 compounds, showed activity in the USP21 counter-screen assay. 16 compounds displayed no activity against USP21 and were selective for PLpro. **b** Screening hit **WEHI-P1** was optimised to **WEHI-P4**. The **WEHI-P1** core structure represents a novel scaffold not seen in any other PLpro inhibitor and exhibits a methoxynaphthyl group bridged by a ketone (in red) to a piperidine with a substituted cyclopentyl group (in blue). Replacement of the ketone to an *O*-methyloxime in **WEHI-P2**

generated activity in the cellular FRET assay (11 μM). Replacing the cyclopentyl with a pendant cyclohexanol and enantiomer separation generated **WEHI-P4** with potent biochemical, cellular and antiviral activity. Compounds were assessed for biochemical (IC$_{50}$, UbRh, Supplementary Fig. 2a) cellular (FRET EC$_{50}$, Supplementary Fig. 2c), and SARS-CoV-2 plaque assay activity (Antiviral EC$_{50}$, Supplementary Fig. 2d), and by SPR ($K_D$, Supplementary Fig. 2g). **c** Crystal structures of PLpro bound to **WEHI-P1** and **WEHI-P4** were determined in different space groups. The BL2 region of PLpro is coloured yellow. The BL2 loop is not involved in **WEHI-P1** binding and participates in a crystal contact (Supplementary Fig. 3a-c, Supplementary Table 1). In **WEHI-P4** the BL2 region is in the 'closed' conformation to encompass the compound. A zoomed-in view of PLpro bound to **WEHI-P4** is shown with key residues labelled. **d** Structure of **WEHI-P4** (top) and GRL0617 (bottom, PDB: 7JIR[34]) with proteins under a semi-transparent surface. The **WEHI-P** series induces a conformational change of PLpro Met208 to expose a pocket not seen in the GRL-0617 bound structure or any other published PLpro inhibitor complex structures. Figure 1a Created in BioRender[92].

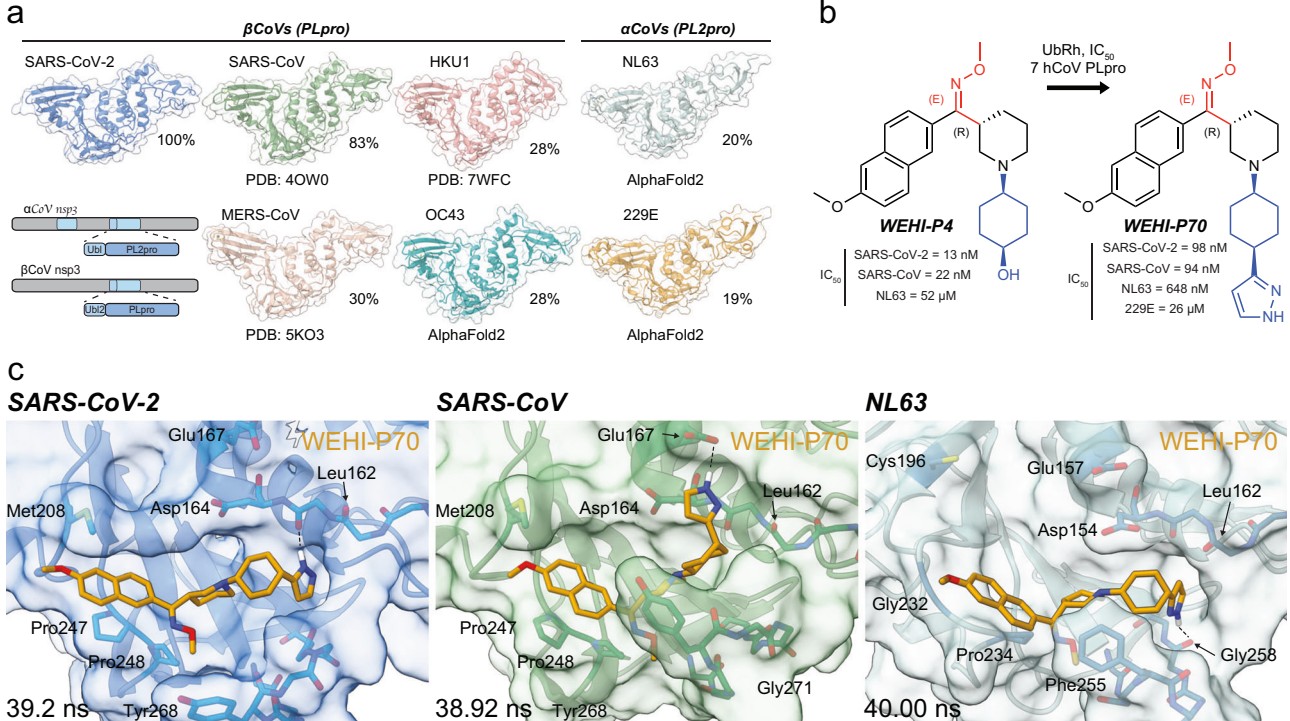

**Fig. 2 | Pan-activity of *WEHI-P* series towards human CoV PLpros. a** SARS-CoV-2 is one of seven coronaviruses pathogenic to humans. βCoVs (including SARS-CoV-2) have one PLpro domain while αCoVs encode two PLpro domains in nsp-3 (indicated in cartoon, with PLpro domains in blue). αCoV PL2pro domains show high similarity to SARS-CoV-2 PLpro, and were recently confirmed to be DUBs[36]. PDB accession codes for each structure are 4OW0 (SARS-CoV[58]), 5KO3 (MERS[91]), 7WFC (HKU1[36]). AlphaFold2 was used for OC43, NL63 and 229E. Amino acid sequence identity relative to SARS-CoV-2 PLpro is indicated as a percentage (%) (see Supplementary Figs. 4b, 5). **b** Biochemical IC$_{50}$s of *WEHI-P4* and *WEHI-P70* against UbRh active PLpros (see Supplementary Fig. 4d, e). **c** 40 ns molecular dynamics simulations docking *WEHI-P70* into indicated CoV PLpro. Key residues are noted. A H-bond formed between the NH at the 2-position on *WEHI-P70* and the backbone carbonyl of Gly258 in NL63 PL2pro appears to stabilise inhibitor binding.

NL63 (IC$_{50}$ 648 nM) and weak activity against 229E (IC$_{50}$ 26 μM) (Fig. 2b, Supplementary Fig. 4d, e).

To understand the molecular basis for a pan-active PLpro inhibitor we turned to molecular dynamics (MD) simulations. We first ran 40 ns MD simulations to dock *WEHI-P4* into our SARS-CoV-2 PLpro complex structure, which produced an almost identical binding mode compared to our experimental structure (Supplementary Fig. 4f). A key difference was in the orientation of the cyclohexanol ring, which in the docking model formed a hydrogen bond with the backbone of Leu162 instead of an interaction with the backbone of Tyr268 as observed in the co-crystal structure. Interestingly, we had seen this cyclohexanol conformation in a 1.88 Å co-crystal structure with *WEHI-P24*, a ketone-compound with cyclohexanol substitution (Supplementary Fig 4f, g, Supplementary Table 1). This suggests that our compounds retain some orientational flexibility that could be exploited.

We next extended the docking studies to further enzymes and compounds. *WEHI-P4* docked similarly into SARS-CoV PLpro, but not into the AlphaFold2 model of NL63, because of unstable interactions around the naphthyl ring. Interestingly, *WEHI-P70* docked with an overall similar binding mode into SARS-CoV-2, SARS-CoV, and NL63 PLpros (Fig. 2c). An additional hydrogen bond between the NH at the 2-position on *WEHI-P70* and the backbone carbonyl of Gly258 in NL63 PL2pro appears to stabilise inhibitor binding to give an increase in activity (Fig. 2c). Together, our work highlights that the *WEHI-P* series has the potential to lead to pan-CoV active antivirals.

## WEHI-P8 is highly efficacious in in vivo models of SARS-CoV-2 disease

Next, in order to enable in vivo studies, we turned our attention to *WEHI-P8*, which features an *O*-ethyloxime (Fig. 3a) and has similar

biochemical activity to *WEHI-P4* (IC$_{50}$ = 12 nM – *WEHI-P8*; IC$_{50}$ = 19 nM – *WEHI-P4*, binding affinity ($K_D$ = 9.0 nM) and cellular activity in a FRET assay (EC$_{50}$ = 298 nM) (Fig. 3a, Supplementary Fig. 6a–c). We further characterised *WEHI-P8* and observed suitable in vitro PK properties, including low hERG binding (IC$_{50}$ = 5.39 μM), and good selectivity against 7 CYPs (IC$_{50}$ = > 20 μM, 1A2, 2B6, 2C8, 2C9, 2C19, 3A4/5; 13.8 μM 2D6). Crucially and unlike currently marketed Mpro inhibitors, *WEHI-P8* shows no time-dependent CYP inhibition against CYP3A4/5 with or without NADPH (IC$_{50}$ > 20 μM) (Fig. 3a, Supplementary Fig. 6d). Moreover, *WEHI-P8* displayed an attractive in vitro PK profile, including low clearance in mouse and human hepatocytes (16.8 and 10.0 μL/min/10$^6$ cells, respectively) and high solubility (Supplementary Fig. 6). Oral administration of *WEHI-P8* at 100 mg/kg in C57BL/6 mice (WT) gave high plasma concentrations over 24 h with an apparent $t_{1/2}$ of 14 h, C$_{max}$ of 7.07 μM and $t_{max}$ of 1 h, suggesting that it had the potential to demonstrate in vivo efficacy (Fig. 3b). To benchmark the SARS-CoV-2 antiviral activity of *WEHI-P8* to the recently published Pfizer PLpro inhibitor compound PF-07957472[22] we performed Vero plaque assays. *WEHI-P8* displayed an antiviral EC$_{50}$ of 360 nM, compared to 460 nM of PF-07957472 (Fig. 3a, Supplementary Fig. 6e, f). Pgp inhibition only slightly improved antiviral efficacy for both *WEHI-P8* (EC$_{50}$ = 290 nM) and PF-07957472 (EC$_{50}$ = 360 nM) (Supplementary Fig. 6e, f). Collectively, this identified *WEHI-P8* as a suitable in vivo candidate.

We previously developed mouse models of SARS-CoV-2 infection to reproduce a spectrum of acute COVID-19 outcomes[38,39]. Animals infected with our SARS-CoV-2 clinical isolate (P2 strain), which naturally infects WT mice due to a N501Y spike mutation, develop mild disease. Serial passaging of the P2 strain in WT mice produced a mouse-adapted virus (P21) that induces a robust inflammatory

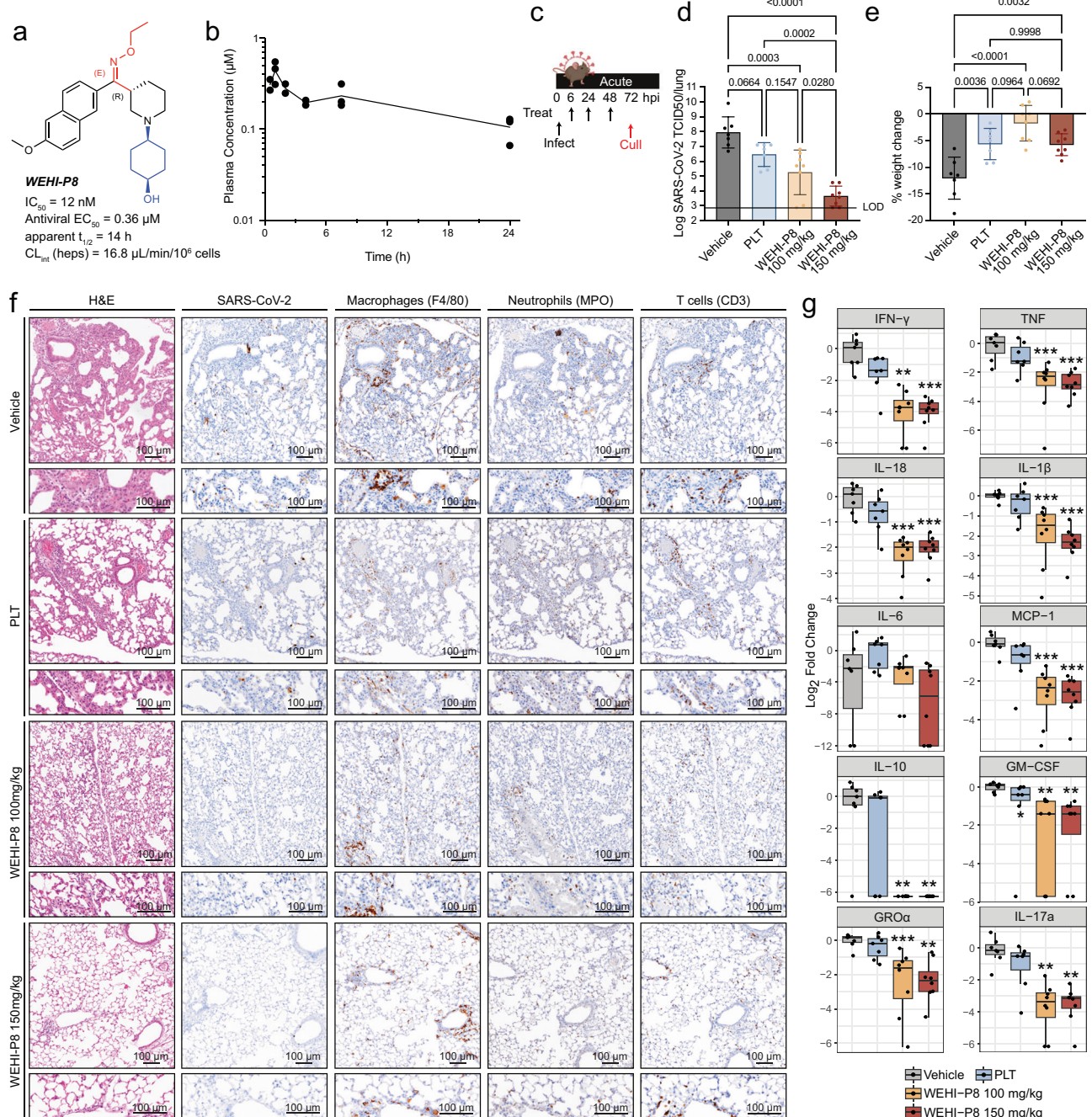

**Fig. 3 | *WEHI-P8* improves disease outcome in an acute mouse model of severe disease. a *WEHI-P8*** was selected for mouse in vivo efficacy due to its favourable ADME properties. **b** Calculated unbound plasma concentration of *WEHI-P8* in male C57BL/6 (WT) mice following oral administration at 100 mg/kg. **c** Schematic showing treatment regime used in **d-g**. Mice were treated at 6 h, 24 h and 48 h with euthanasia performed at 72 h post-infection. WT 7-9 week-old mice were infected with SARS-CoV-2 P21 (see Supplementary Fig. 7a) and treated with either vehicle, PLT (Paxlovid-like treatment: 56 mg/kg nirmatrelvir, 19 mg/kg ritonavir), or *WEHI-P8* (100 mg/kg or 150 mg/kg) (see schematic and **Methods d,e** At 3 days port-infection (dpi), mice were monitored for **d** viral burden and **e** percent weight change compared to initial weight; $n_{vehicle} = 7$, $n_{PLT} = 7$, $n_{P8-100} = 8$, $n_{P8-150} = 8$ mice per group. Mean ± SD. **f** Haematoxylin and eosin (H&E) and immunohistochemistry (IHC) stained lungs are shown. Markers used for each cell type are indicated in brackets and images are representative of 4 animals per condition. Scale bars = 100 μm. **g** Levels of cytokines and chemokines measured by ELISA of lung homogenates from mice infected with SARS-CoV-2 P21; $n_{vehicle} = 7$, $n_{PLT} = 7$, $n_{P8-100} = 8$, $n_{P8-150} = 8$ mice per group; boxplots depict the median and interquartile range (IQR). Whiskers extend to the furthest data point within 1.5 times the IQR from each box end. *P*-values are indicated above each group and were determined by **e** one-way ANOVA with Tukey's multiple comparisons tests **d** after $log_{10}$ transformation and **g** Two-sided wilcoxon rank-sum test, with Bonferroni adjustment for multiple comparisons; **p < 0.01, ***p < 0.001. Exact *P*-values for Fig. 3g are provided in the Source Data file. Source data are provided as a Source Data file. Figure 3c Partially created in BioRender[93].

immune response in the host, resulting in severe disease (Supplementary Fig. 7a). We utilised these models to perform a pre-clinical assessment of *WEHI-P8* in improving acute disease outcomes in both mild and severe disease. To define an appropriate in vivo dosing regimen, we conducted pilot experiments using our mild disease model (SARS-CoV-2 P2 strain), treating at -2 h pre-, 6 h, and 24 h post-infection (Supplementary Fig. 7b). Nirmatrelvir (150 mg/kg) and ritonavir (19 mg/kg) alone showed no antiviral activity in vivo

(Supplementary Fig. 7c), which we attributed to the absence of co-dosing, a requirement for efficacy in humans. Consequently, we calculated the mouse-equivalent dose of Paxlovid prescribed for COVID-19 patients, based on a human body weight of 65 kg (300 mg nirmatrelvir, 100 mg ritonavir) and a literature-derived conversion factor of 12.3[40]. The resulting PLT dose for mice was 57 mg/kg nirmatrelvir and 19 mg/kg ritonavir, which we used throughout this study as a control. This PLT regimen demonstrated antiviral efficacy in inhibiting P2 virus replication, comparable to that observed with **WEHI-P8** administered at 150 mg/kg using the same dosing schedule (Supplementary Fig. 7c). Furthermore, our data suggest that the antiviral activity of **WEHI-P8** is independent of ritonavir and cannot be further enhanced by concurrent ritonavir co-treatment (Supplementary Fig. 7c). Notably, initiating treatment after infection with our clinical isolate P2 (6 h, 24 h, and 48 h) did not change the antiviral efficacy of PLT and **WEHI-P8** (Supplementary Fig. 7d).

### WEHI-P8 prevents severe outcomes in a model of severe SARS-CoV-2 disease

Based on the experimental findings that both PLT and **WEHI-P8** were efficacious in our mild mouse model, we utilised the same treatment regimen in a severe model of SARS-CoV-2 infection. To analyse the effect of antiviral treatment on the host inflammatory response, we treated animals after infection (6 h, 24 h, and 48 h) with our mouse-adapted SARS-CoV-2 strain P21 (Fig. 3c). P21 infection is characterised by increased lung viral titres, weight loss, pronounced cytokine response, lung immune cell influx and overt inflammation[38]. SARS-CoV-2 P21 is highly pathogenic to mice, with animals older than 9 weeks reaching ethical endpoint requiring euthanasia ( > 20% weight loss) (Supplementary Fig. 7e). Notably, **WEHI-P8** was more effective than PLT at reducing viral burdens in 6–7-week-old mice infected with SARS-CoV-2 P21 (Fig. 3d). While both treatments were able to reduce disease severity by rescuing mice from weight loss (Fig. 3e and Supplementary Fig. 7f), H&E staining revealed that PLT treated mice exhibited mild to moderate multifocal inflammation, primarily localised to peribronchiolar and perivascular regions of the lungs, with occasional alveolar collapse and mild haemorrhagic foci. In contrast, animals treated with 100 mg/kg of **WEHI-P8** displayed only mild inflammation, which was further reduced in the group treated with 150 mg/kg, where most areas showed minimal to unremarkable histological changes (Fig. 3f). Immunohistochemistry (IHC) for immune cell subsets highlighted that **WEHI-P8** was able to significantly reduce the numbers of macrophages and neutrophils at 100 mg/kg, while 150 mg/kg also reduced the number of T cells. PLT reduced the number of macrophages, but not neutrophils or T cells (Fig. 3f and Supplementary Fig. 7g). Cytokine and chemokine profiling in the lungs found that while PLT led to a significant reduction in GM-CSF levels, 100 mg/kg of **WEHI-P8** was enough to significantly reduce the levels of a wider range of pro-inflammatory cytokines and chemokines, including those commonly associated with increased disease severity (IFN-γ, IL-1β and TNF)[41–43] (Fig. 3g, Supplementary Fig. 7h). Notably, administering **WEHI-P8** as a pre-infection regimen (2 h before infection, followed by doses at 6 h and 24 h post-infection, Supplementary Fig. 8) was no more effective than initiating treatment after infection (Fig. 3). In contrast, PLT's antiviral efficacy was contingent on early dosing (Supplementary Fig. 8).

### Recapitulating key hallmarks of PASC in a mouse model

There are currently no treatments for PASC, and robust pre-clinical mouse models that reflect the majority of human symptoms are needed to test the efficacy of potential treatment strategies[8,14]. The recently reported mouse models of PASC[44–46] rely on a genetically engineered viral strain, modified to enhance infectivity in mice[47] and are not derived from human isolates. In contrast, our acute model of severe disease, is derived from an Australian circulating variant.

Importantly, when mice cleared SARS-CoV-2 P21 by 7 dpi (virus titres below lower limit of detection (LOD)) (Supplementary Fig. 9a), lungs of infected animals still show histological signs of disease[38]. This prompted us to investigate the long-term sequelae of disease in our model. To this end, we infected different cohorts of mice aged 9–14 weeks and monitored them for weight loss during the acute phase of infection, before various organs were collected between 1- and 3-months post-infection (mpi) (Fig. 4a). In adult animals older than 10 weeks, SARS-CoV-2 P21 infection causes more severe disease, with over 50% requiring euthanasia due to significant weight loss or extreme signs of disease (Fig. 4b, c). At 1 mpi, histology revealed that mock animals were histologically unremarkable, but importantly, most animals that survived SARS-CoV-2 acute disease yet retained distinctive markers indicative of PASC. In recovered animals, moderate to severe haemorrhage was found in the lumen of bronchioles and throughout the alveoli (Fig. 4d). At 3 mpi, the alveolar haemorrhage remained while a proportion of recovered animals also presented with type II pneumocyte hypertrophy hyperplasia, suggesting a repair response to lung damage was occurring (Fig. 4d). Peribronchiolar and perivascular immune infiltrates were present at both 1 and 3 mpi with IHC indicating this was composed primarily of macrophages and T cells (Fig. 4d). Quantification of haemorrhage and immune cell infiltration revealed that inflammatory cells decrease at 3 mpi, while signs of haemorrhage persist until this later time point, indicating a prolonged vascular pathology (Fig. 4e). To investigate signs of fibrosis, we analysed collagen deposition over time. At 1 mpi, the extent of collagen-positive areas in the lungs was comparable to that observed in mock animals. However, by 3 mpi, previously infected animals exhibited a significant increase in lung collagen, indicative of a fibrotic remodelling (Fig. 4d, f). These phenotypes are remarkably similar to long-term pathophysiological manifestations found in the lungs of patients with PASC[48].

To interrogate potential molecular drivers of lung pathogenesis in the PASC model, we compared the bulk lung proteomes of SARS-CoV-2 infected and mock animals at 45 dpi. Animals that recovered from severe disease showed 36 significantly differentially expressed proteins compared to mock (Supplementary Fig. 9d). 1D annotations revealed an increase in pathways responsible for T-cell mediated immune responses, MHCII expression and antigen presentation, collectively indicating persistent immune activation in the lungs that could underpin long-term sequelae after SARS-CoV-2 infection (Fig. 4g). Additionally, dysregulated Laminin 5 complex and surfactant homoeostasis pathways suggest a damage repair response is occurring in the lung (Fig. 4g).

### Heart, brain and gut dysfunction in animals recovered from SARS-CoV-2 P21

In humans, SARS-CoV-2 infection can lead to a range of long-term clinical symptoms beyond the lung, including: cardiac dysfunction and enlarged ventricles of the heart[49]; gut dysfunction[50]; and cognitive impairment, commonly termed 'brain fog'[8]. We next set out to understand whether and how our PASC mouse model reflects these human disease hallmarks.

H&E staining of the hearts from recovered animals at 1 mpi showed significantly enlarged right ventricles compared to mock controls (Fig. 4h, i), suggesting on-going and persisting heart alterations, which persist at 3 mpi. Similarly, we H&E stained and scored sections of the small and large intestines based on epithelial damage, hyperproliferation of crypts and crypt loss (Fig. 4j). Our analysis indicated that at both 1 and 3 mpi, mice previously infected with SARS-CoV-2 displayed a higher overall histological score, indicating marked gut pathology compared to control mice (Fig. 4k).

Finally, we assessed whether mice recovered from acute SARS-CoV-2 driven disease also displayed signs of neurological abnormalities. Microglia are a key cell type for mediating immune responses in

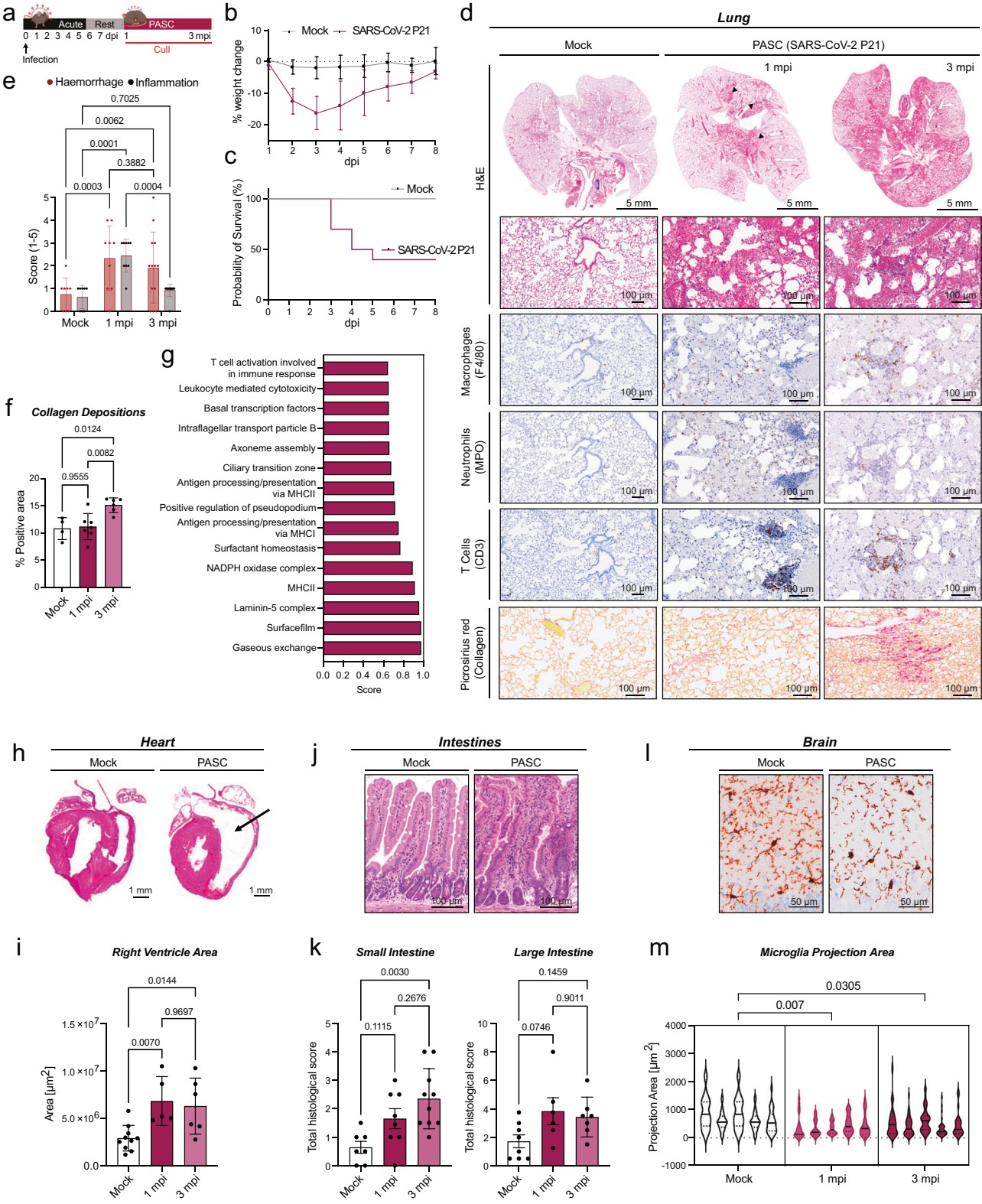

the brain, and dysregulated microglia function has been linked to acute and long term COVID-19 outcomes[51,52]. Under resting conditions, microglia have small cell bodies and numerous long branching cell protrusions, making up a ramified morphology. Upon activation, the protrusions retract and thicken, decreasing the projection area of these cells, and they begin secreting cytokines and radical species[53]. We stained fixed brain sections with the microglial marker IBA-1 and quantified cell activation by analysing the extent of microglial

ramification (length of projections). While in mock-animals most cells showed a ramified morphology, SARS-CoV-2 recovered animals at 1 mpi showed a significant reduction in the ramification (projection area) of microglial cells (Fig. 4i, m) indicative of neuroinflammation. Strikingly, infection of WT mice with our SARS-CoV-2 P21 strain recapitulates various PASC symptoms across a range of organs, facilitating studies to understand the full range of long COVID pathology as well as the efficacy of intervention.

**Fig. 4 | A PASC mouse model. a** Schematic showing time points used to analyse long-term SARS-CoV-2 driven disease (PASC). Mice were infected intranasally with SARS-CoV-2 P21 and euthanised between 1 and 3 months post-infection (mpi). **b−m** 9-14 week-old mice were challenged intranasally with either mock (DMEM only) or P21 and monitored daily for **b** percent weight change relative to initial weight and **c** percent of animals reaching humane endpoint requiring euthanasia (results are representative of 6 independent experiments; $n_{mock} = 11$, $n_{pasc} = 10$ animals per group, mean ± SD). Organs were collected for: **d** Haematoxylin and eosin (H&E) and immunohistochemistry (IHC) staining of fixed lungs. Markers used for each cell type are indicated in brackets and images are representative from 5 animals per group. Scale bar (top) = 5 mm and (bottom) = 100 μm. **e** Blinded scoring of H&E-stained lung sections was performed. The lung was assessed for the presence of inflammatory foci and haemorrhage, with pathology scored on a scale from 0 to 5. A score of 0 indicated no detectable pathology, while a score of 5 represented extensive pathology; $n_{mock} = 8$, $n_{1mpi} = 9$, $n_{3mpi} = 13$ animals per group; Mean values ± SEM. **f** Positive area of Picrosirius red staining was quantified relative to total lung area; $n_{mock} = 4$, $n_{1mpi} = 7$, $n_{3mpi} = 6$ animals per group. **g** Lungs of mock

and PASC animals were taken at 45 days post-infection (dpi) for bulk proteomics analysis. 1D annotation enrichment analysis of proteins changing in lungs post-infection compared to mock is shown (significance was set to Benjamini Hochberg FDR < 0.02). $n_{mock} = 4$, $n_{pasc} = 5$ animals per group. **h** Histological analysis of H&E-stained hearts. Scale bar = 1 mm. **i** Quantification of right ventricle area ($n_{mock} = 8$, $n_{1mpi} = 5$, $n_{3mpi} = 6$ animals per group.; Mean ± SEM). **j** Histological analysis of H&E-stained intestines. **k** Total histological score of the small ($n_{mock} = 7$, $n_{1mpi} = 8$, $n_{3mpi} = 10$ animals per group) and large intestines ($n_{mock} = 8$ $n_{1mpi} = 6$, $n_{3mpi} = 7$ animals per group; mean values ± SEM). **l** IHC of fixed brains at 45 dpi, stained with IBA-1 (microglia). Scale bar = 50 μm. **m** Quantification of the projection area of cells positive for IBA-1 staining in the hippocampus ($n = 5$ animals per group; >18 cells were counted per mouse; violin plots show median and quartiles). *P*-values are indicated above the graph and were determined by **e** Mixed effect analysis with Tukey's multiple comparison tests (**f, i, k**) One-way ANOVA with Tukey's multiple comparisons tests **f,l,k** and **m** nested one-way ANOVA. Source data are provided as a Source Data file. Figure 4a Partially created in BioRender[94].

Ageing is associated with heightened susceptibility to severe outcomes following SARS-CoV-2 infection[54]. To elucidate age-related disease pathology in our model, we analysed the long-term effects of infection in the lungs of aged C57BL/6 mice (>6 months old) infected with SARS-CoV-2 P21 at 3 mpi. Given that aged mice typically succumb to infection around 5 dpi[38] when challenged with standard doses, we administered a lower inoculum (200 TCID50) of P21 to extend survival beyond the acute phase of disease, enabling the study of later pathological events. With this dose, all animals survive acute phase of infection, losing a maximum of 16% of body weight (Supplementary Fig. 9c, d). We observed that pathology differed in aged animals compared to adult cohorts, with aged animals more likely to develop multifocal peribronchial and perivascular lymphocytic aggregates, which often assumed an organised pattern similar to BALT-like structures (Supplementary Fig. 9e, f). These immune cell foci assumed an organised structures, mainly composed of macrophages and T cells, but not neutrophils (Supplementary Fig. 9e). Similar to young mice, aged animals that recovered from a low dose of SARS-CoV-2 P21, showed a significant reduction in the ramification (projection area) of microglial cells in the brain (Supplementary Fig. 9g).

## WEHI-P8 protects mice from SARS-CoV-2 induced long-term symptoms in lung and brain

It remains unclear whether antiviral treatments can alleviate the long-term symptoms seen in patients suffering from PASC[6,14]. Our findings demonstrate that prophylactic administration of ***WEHI-P8*** (Supplementary Fig. 8a) did not confer a significant benefit compared to post-infection treatment initiation (Fig. 3d). In contrast, PLT was only effective against SARS-CoV-2 P21 when administered prior to infection (Supplementary Fig. 8a). Guided by these observations and to enable direct comparison, we treated adult mice with PLT or ***WEHI-P8*** at 2 h pre-infection, and 6 h and 24 h post-infection with P21 virus, and rested animals for 1 month before assessment of PASC symptoms (Fig. 5a). In our PASC model, over 50% of animals succumbed to acute infection due to significant weight loss (Fig. 4b, c). To ensure sufficient numbers for assessing PASC outcomes following drug treatment, supportive care was provided, including saline administration and mashed food supplementation for animals with >15% body weight loss. Consistent with our results during acute infection, both treatments effectively prevented infection-induced weight loss, with saline support required only for vehicle-treated animals (Fig. 5b). While a subset of untreated animals required euthanasia due to severe weight loss and low body condition scores, all treated animals survived the acute phase of infection (Fig. 5c). H&E staining of the lungs highlighted that vehicle-treated animals displayed many of the pathology markers as described for the PASC mice above, including haemorrhage and inflammatory

foci (Figs. 4d, 5d). PLT-treated animals showed a similar sustained moderate to severe haemorrhage in the lung and presence of immune cell infiltrates (Fig. 5d, e). Strikingly, animals treated with ***WEHI-P8*** showed reduced signs of haemorrhage and immune cell infiltrates, with lungs of most animals looking histologically unremarkable. This indicated that while PLT was unable to prevent PASC-like lung symptoms in the dosing regimen given, our efficacious PLpro inhibitor rescued animals of otherwise expected long-term lung abnormalities following SARS-CoV-2 infection (Fig. 5d, e). Interestingly, not all PASC hallmarks were equally affected by ***WEHI-P8*** treatment. In the brain, neuroinflammation as measured by microglial activation (IBA-1 staining, see above), was significantly reduced with ***WEHI-P8***, but not with PLT (Fig. 5f). In contrast, analysis of right ventricle area showed no significant differences between vehicle-treated or antiviral-treated groups (Supplementary Fig. 9h). Similarly, although a trend was observed, increased gut inflammation could not be rescued by neither PLT or ***WEHI-P8*** treatment (Supplementary Fig. 9i).

Recently, mice recovering from SARS-CoV-2 were shown to exhibit long-term neurological effects manifesting as reduced performance in a range of behavioural, memory and cognitive tests[14]. We performed behavioural studies using the Novel Object Recognition Test (NORT)[55]. In NORT, rodents initially explore two identical objects. During a subsequent trial, one of the familiar objects is replaced with a novel object. Rodents naturally prefer novelty, and a mouse with intact recognition memory will spend more time exploring the novel object (Supplementary Fig. 9j). In our PASC infection model, vehicle- and PLT-treated mice exhibited a similar reduction in exploratory behaviour, while ***WEHI-P8***-treated animals demonstrated a significantly higher preference for the novel object compared to vehicle and PLT-treated groups (Fig. 5g). Interestingly, sex-specific analysis revealed that infection more adversely affected female than male behaviour. This impairment was partially mitigated by ***WEHI-P8*** treatment, whereas PLT treatment showed no benefit (Supplementary Fig. 9k). To further examine behavioural differences between males and females, we analysed the recognition index, defined as the proportion of time spent exploring the novel object relative to the total exploration time. This metric did not reveal significant sex- or drug dependent differences upon infection (Fig. 5h). However, the overall distance travelled during behavioural tests was significantly reduced in infected females, but not males. Notably, this effect was rescued by ***WEHI-P8*** treatment but not by PLT (Fig. 5i). Overall, our findings suggest that female animals are more susceptible to PASC-related cognitive symptoms, which appears to be a further similarity to humans[56]. These deficits can be ameliorated by ***WEHI-P8*** treatment, but not PLT. Collectively, our data highlight the potential of antivirals in effectively preventing PASC-associated symptoms.

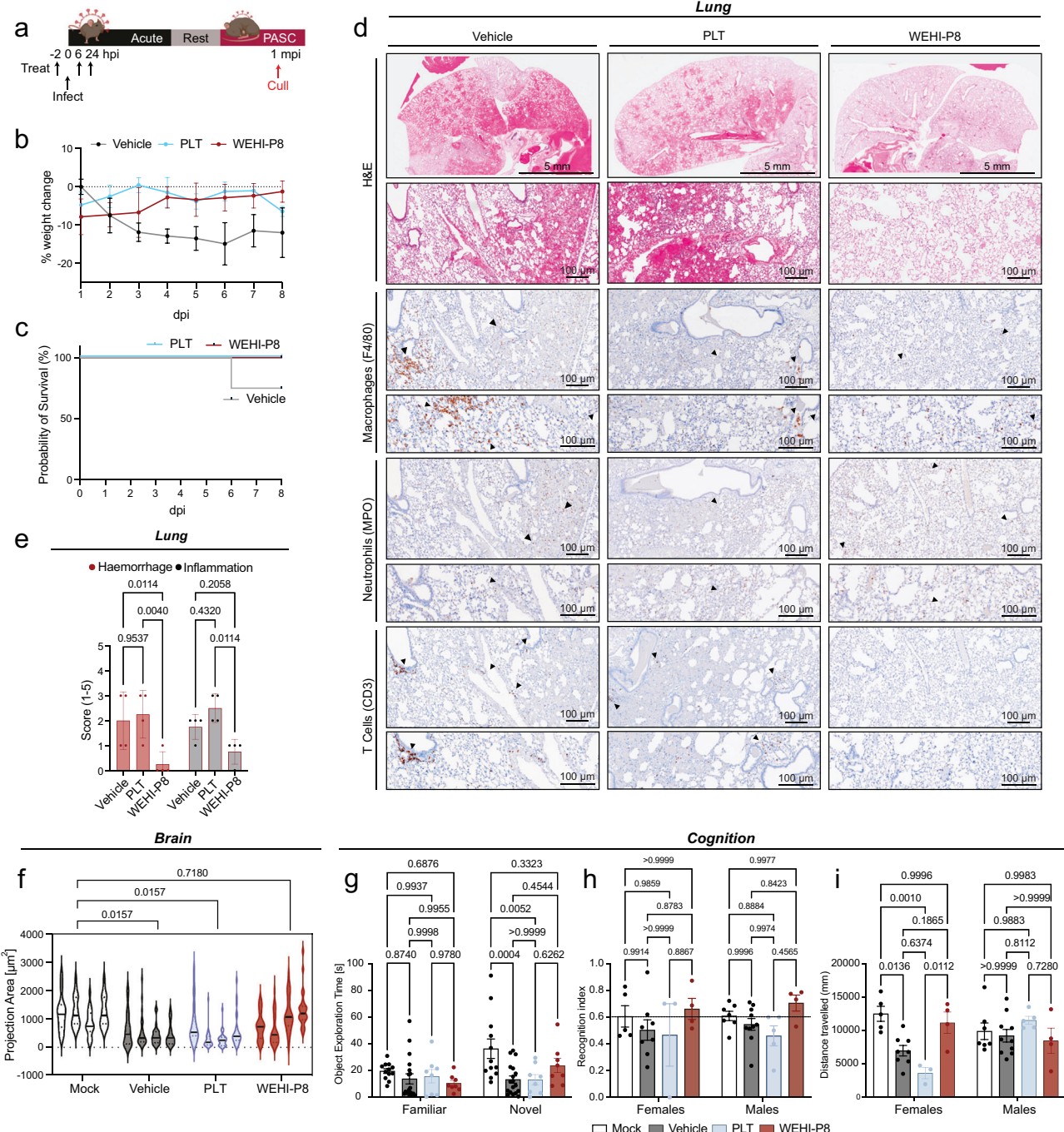

**Fig. 5 | WEHI-P8 significantly reduces post-acute manifestations of disease in lungs and brain. a** Schematic showing treatment regime: Mice were treated at -2 h pre- intranasal infection with SARS-CoV-2 P21, treated again at 6 h and 24 h post-infection with either vehicle, PLT (Paxlovid-like treatment, 56 mg/kg nirmatrelvir, 19 mg/kg ritonavir) or **WEHI-P8** (150 mg/kg), rested and euthanised for downstream analysis at 30 days post-infection (dpi). **b**−**i** 11–12-week-old P21 infected mice were treated and monitored daily for **b** percent weight change relative to initial weight (mean values ± SEM) and (**c**) percent of animals reaching humane endpoint requiring euthanasia (results are representative of 2 independent experiments; $n_{veh}$ = 11, $n_{PLT}$ = 8, $n_{PS}$ = 8 mice per group). Organs were collected for: **d** Haematoxylin and eosin (H&E) and immunohistochemistry (IHC) staining of lungs. Markers used for each cell type are indicated in brackets and images are representative of 4 animals per group. Scale bar (top) = 5 mm and (bottom) = 100 μm. **e** Blinded scoring of H&E-stained lung sections. The entire lung was assessed for the presence of inflammatory foci and haemorrhage, with pathology

scored on a scale from 0 to 5. A score of 0 indicated no detectable pathology, while a score of 5 represented extensive pathology affecting the entire lung; $n$ = 4 animals per group; Mean values ± SEM. **f** Quantification of the projection area of cells positive for IBA-1 (microglia) staining in the hippocampus ($n$ = 4 animals per group; >18 cells per mouse were counted; violin plots show median and quartiles). **g**−**i** Novel object recognition test. **g** time spent exploring the familiar and novel objects is shown. **h** The recognition index was calculated as a proportion of the time exploring the novel object over the total time spent exploring both objects (Chi-squared (threshold 60%) = 0.51). **i** Total distance (mm) travelled by mice during the novel object recognition test. $n_{mock}$ = 12, $n_{veh}$ = 18, $n_{PLT}$ = 8, $n_{PS}$ = 8 mice per group; Mean values ± SEM. In all cases $p$-values are indicated above the graph and were determined by (**e, g, h, i**) Two-way ANOVA with Sidak's multiple comparison tests and **f** nested one-way ANOVA with Tukey's multiple comparisons test. Source data are provided as a Source Data file. Figure 5a Partially created in BioRender[95].

## Discussion

The COVID pandemic has transitioned into an endemic phase in which millions continue to suffer from the acute effects of the disease, and increasingly from PASC related complications. New antivirals that target the orthogonal essential protease PLpro in SARS-CoV-2 will provide new and better treatment options, either alone or in combination with Mpro inhibitors, for COVID-19. Molecules that can pivot to a pandemic preparedness approach with broad-spectrum PLpro activity may be useful for the next novel coronavirus outbreak.

In this study, we introduce the *WEHI-P* series, a novel class of PLpro inhibitors that target coronaviruses through a binding mode distinct from known scaffolds and show potential as pan-coronavirus antivirals. Notably, *WEHI-P8* demonstrated oral efficacy in vivo, including in a severe SARS-CoV-2 disease model, where it outperformed PLT in terms of both viral inhibition and reduction of the inflammatory response in the human-equivalent regimen given. Whether this difference stems from its reduced in vivo efficacy in lowering viral burden in our model of severe disease compared to *WEHI-P8* or is a result the distinct biological roles of Mpro and PLpro remains an open question. Virus and disease kinetics in mouse models differ to human COVID-19 disease. While viral replication in mice occurs rapidly and peaks in the lungs of infected animals after 24 h, disease symptoms (weight loss) only peaks at 3-4 dpi. In contrast, the onset of clinical symptoms in humans coincides with peak viral burdens[57]. While later treatment initiation might be clinically relevant for individuals who do not seek medical attention immediately, our study aimed to evaluate the impact of an optimal, early antiviral intervention that is analogous to the currently most beneficial strategy in human COVID-19 infection. Nevertheless, side-by side comparisons of PLpro and Mpro inhibition in other mouse models, using different virus strains and across extended treatment regimens may be required to enable generalisability and enable the translation of these results to the clinic.

Furthermore, we established an in vivo model of PASC using a clinically relevant viral isolate that induces long-term sequelae across multiple organs, including behavioural changes, which are more pronounced in female animals. This model provides a valuable platform for testing preventive and curative interventions and enables future studies utilising gene-targeted animals to elucidate the molecular pathways and mechanisms underlying long-term pathology.

Our findings demonstrate that early PLpro inhibition, with the novel *WEHI-P8* compound, not only proves highly effective in acute COVID-19 but also prevents the development of long COVID symptoms when given prophylactically. This proof-of-principle study offers a promising outlook for the development of effective strategies targeting PLpro to prevent or treat long COVID, and it will be exciting to see whether these preclinical results translate to clinical settings where antiviral treatment is initiated after the onset of symptoms of acute disease. Further investigation on the WEHI-P series will be required to progress these compounds towards clinical translation. This includes a more detailed interrogation of longitudinal disease outcomes upon PLpro inhibition in multiple organ systems during long COVID, specifically with antivirals given at later stages of infection. Human-relevant dosing with extended and optimised treatment regimens will have to be established before clinical trials can be initiated. Collectively, our results provide further evidence that PLpro is a promising antiviral drug target for COVID-19 with the potential to alleviate long COVID outcomes, and that PLpro inhibitor compounds can be an important asset for pandemic preparedness.

## Methods

### Protein biochemistry and structural biology

**Reagents.** Ubiquitin Rhodamine110 (UbRh, UbiQ Bio, UbiQ-002), isopropyl ß-D-1-thiogalactopyranoside (IPTG, Gold-Bio #I2481C100), β-mercaptoethanol (Sigma M3148), imidazole (Sigma 56749), lysozyme (Glentham Life Science GE8228)

DNase I (Roche 11284932001), tris(2-carboxyethyl)phosphine, (TCEP, GoldBio TCEP25), Bovine Serum Albumin (Sigma, A2153), Triton-X-100 (Sigma 9002-93-1)

L-Glutathione-Reduced (L-GSH, Sigma G4251), DMSO (Sigma-Aldrich, #472301-100 ML), doxycycline (dox, #D5207, Sigma Aldrich), tris(hydroxymethyl)aminomethane (Tris, Sigma Aldrich 9210-OP), L-glutathione (GSH, Sigma Aldrich 1294820), 4-(2-hydroxyethyl) piperazine-1-ethanesulfonic acid (HEPES Sigma Aldrich, H3375). Compound **5c** was synthesised as previously[58].

**Molecular biology.** Bacterial pOPIN-B expression vectors[59] for SARS-CoV-2 PLpro (amino acids (aa) 1564-1878 of polyprotein 1ab, GenBank: QHD43415, with aa E1564 designated as residue 1), were reported previously[18]. SARS-CoV PLpro$^{WT}$ (aa 1541-1855 of polyprotein 1ab, RefSeq: NP_828849.7), MERS-CoV PLpro$^{WT}$ (aa 1482-1803 of polyprotein 1ab, RefSeq: YP_009047202), HKU1-CoV PLpro$^{WT}$ (aa 1648-1958 of polyprotein 1ab, RefSeq: YP_009944268), OC43-CoV PLpro$^{WT}$ (aa 1561-1872 of polyprotein 1ab, RefSeq: AY391777), 229E-CoV PL2pro$^{WT}$ (aa 1599-1905 of polyprotein 1ab, RefSeq: NP_073549), NL63-CoV PL2pro$^{WT}$ (aa 1578-1876 of polyprotein 1ab, RefSeq: YP_003766) were codon optimised for bacterial expression, synthesised (Integrated DNA Technologies) and cloned into pOPIN-B (pOPIN-S for OC43) digested with KpnI and HindIII using In-Fusion™ HD cloning (Takara Clontech). The SARS-CoV-2 PLpro BL2 mutant (SARS-CoV-2 PLpro$^{BL}$) was generated as described previously[27]. For SPR, constructs were ordered with an N-terminal AviTag™ (GLNDIFEAQKIEWHE) and cloned as above. A GSS or GSSG linker was placed preceding the 3 C cleavage site or protein CDS respectively. For crystallography, we matched a construct used previously[27,34], which has a 1-aa shorter SARS-CoV-2 PLpro sequence (aa 1564-1878) preceded by a Ser-Asn-Ala sequence and includes a catalytic Cys111 mutation to Ser (SARS-CoV-2 PLpro$^{C111S}$). The coding sequence was cloned into pOPIN-S which features a His-SUMO-tag. SUMO protease (SENP1) was produced as per literature[60].

**Protein purification.** All protein expression vectors were transformed into *E. coli* Rosetta™ 2(DE3) competent cells (Novagen) and bacterial cells were grown in 2xYT medium at 37 °C. At $OD_{600} = 0.8$ the temperature was reduced to 18 °C and expression was induced with 0.3 mM IPTG. Cells were harvested 16 h post induction and stored at -80 °C until purification. For purification, cells were resuspended in lysis buffer/Buffer A (50 mM Tris pH 7.5, 500 mM NaCl, 5 mM b-ME, 10 mM imidazole) supplemented with lysozyme (2 mg/mL), DNaseI (100 μg/mL), $MgCl_2$ (5 mM) and cOmplete EDTA-free protease inhibitor cocktail tablets (Roche) and lysed by sonication. Lysates were cleared by centrifugation at 40,000 g for 30 min at 4 °C. The clarified lysate was filtered through a 0.45 μM syringe filter and His-tagged proteins were captured using a HisTrap HP column (5 mL, Cytvia). The captured protein was washed with 10 CV of 30 mM imidazole wash buffer (Buffer A + 10% (v/v) Buffer B) and eluted using five column volumes of 100% Buffer B (Buffer A + 300 mM Imidazole). Pooled fractions were desalted into 100% Buffer A using a HiPrep 26/10 Desalting column (Cytiva) and then supplemented with His-3C or His-SENP1 protease for His-tag and His-SUMO tag cleavage respectively. Following overnight incubation at 4 °C, the cleaved His-tag, His-SUMO tag and His-tagged proteases were captured using a HisTrap HP column (5 mL, Cytiva). The extracted PLpro found in the flow-through was further purified by size exclusion chromatography using a HiLoad 16/600 Superdex 75 pg column (Cytiva) equilibrated with storage buffer (20 mM Tris pH 7.5, 150 mM NaCl, 1 mM TCEP).

For HTS, SARS-CoV-2 PLpro$^{WT}$ was purified as above. For SPR storage buffer, 20 mM Tris pH 7.5 was replaced with 10 mM HEPES pH 7.5, for crystallisation, 150 mM NaCl was replaced with 50 mM NaCl. Protein samples were concentrated, and flash frozen in liquid nitrogen and stored at −80 °C.

**PL1pro/PL2pro activity assay.** Activity assays were performed as described previously[18]. In short, SARS-2-CoV PLpro activity was monitored in a fluorescence intensity assay using the substrate Ub-Rhodamine 110 (UbRh), that upon cleavage becomes fluorescent. The assay buffer contained 20 mM Tris (pH 8), 1 mM TCEP, 0.03% BSA (w/v) and 0.01% (v/v) Triton-X. Experiments were performed in 1536-well black non-binding plates (Greiner 782900) with a final reaction volume of 6 μL. SARS-2-CoV PLpro enzyme was added to the plates (50 nM or 5 nM) and incubated at ambient temperature for 10 min. UbRh (final concentration 100 nM) was added to start the reaction and incubated for 12 min (50 nM PLpro), or 2 h (5 nM PLpro), at room temperature. For endpoint assays the reaction was stopped by addition of citric acid (1 μL) at a final concentration of 10 mM. All additions were performed using the CERTUS FLEX (v2.0.1, Gyger). The reaction was monitored by an increase in fluorescence (excitation 485 nm and emission 520 nm) on a PHERAstar® (v5.41, BMG Labtech) using the FI 485 520 optic module. Data was normalised to 1% (v/v) DMSO (negative control, 0% inhibition) and 100 μM **Sc**[58] (positive control, 100% inhibition).

**High throughput screen.** A total of 412,644 compounds (in-house library) were screened using the PLpro activity assay. Assay-ready plates were prepared at Compounds Australia. Compounds were dry spotted onto 1536-well non-binding black plates (Greiner 782900) to a final concentration of 29.17 μM in 2% (v/v) DMSO. Stock concentrations of compounds were either 10 mM or 5 mM. Reagents were dispensed using the CERTUS FLEX (v2.0.1, Gyger). Microplates were centrifuged using the Microplate Centrifuge (Agilent) and read on the PHERAstar® (v5.41, BMG Labtech) using the FI 485 520 optic module.

Data was normalised to 2% (v/v) DMSO (negative control, 0% inhibition) and 100 μM **Compound Sc**[58] (positive control, 100% inhibition). Screen assay quality was monitored by calculation of robust Z' by the following formula where (+) denotes the positive controls (low signal), (-) denotes the negative controls (high signal) and MAD is the median absolute deviation: robust $Z' = 1 - (3*(MAD_- + MAD_+) / abs(median_- - median_+))$ where $MAD = 1.4826 * median(abs(x - median(x)))$. Plates were excluded from analysis if robust $Z' < 0.5$. Hits were selected as >3* SD over the median of the negative control. For 36 plates, hits were selected as >1.5 SD over the median of the negative control due to an over dispense of DMSO (4% (v/v) DMSO final) during assay-ready plate preparation.

To determine the potency of the inhibitors, a series of 10-pt, 1:2 serial dilutions was performed from the highest starting concentration of 100 μM. The 10-point titration curves were fitted with a 4-parameter logistic nonlinear regression model and the IC50 reported is the inflection point of the curve. Data was analysed in TIBCO Spotfire® 7.11.2.

**Counter screen.** To confirm that the compounds were specifically inhibiting SARS-2-CoV PLpro rather than interfering with the fluorescence readout, human USP21 was used as the counter screen assay as previously described[18]. The same buffer, reagent dispenser and plate reader as in the PLpro assay was used. USP21 enzyme (final concentration 5 nM) was added to the plates and incubated at room temperature for 10 min. UbRh (final concentration 100 nM) was added to start the reaction and incubated for 2 min at room temperature. Reaction was stopped by the addition of citric acid (1 μL) at a final concentration of 10 mM. A series of 10-pt, 1:2 serial dilutions was performed from the highest starting concentration of 100 μM. The 10-point titration curves were fitted with a 4-parameter logistic nonlinear regression model and the IC50 reported is the inflection point of the curve. Data was analysed in TIBCO Spotfire® 7.11.2.

**Comparative analysis of PLpro variants and human DUBs against inhibitors.** Activity and inhibition of PL1pro and PL2pro from diverse viruses, as well as several human DUBs, were tested in PLpro activity assay as described, except that 1 mM GSH was used in place of 1 mM TCEP. The final concentration of UbRh was 100 nM, except for ATXN3 where it was adjusted to 2000 nM to account for low activity of the enzyme.

Experiments were performed in 384-well black non-binding plates (Greiner 784900 or Aurora ABA000000A) with a final reaction volume of 6 μL. A series of 10-pt, 1:3 serial dilutions was performed on test compounds using the Echo® Acoustic Dispenser (LabCyte) with the highest starting concentration of 100 μM of compounds. 5 μL of enzyme was added to the assay-ready plates and incubated for 10 min. UbRh was added to start the reaction and incubated for the required incubation time at room temperature. For endpoint assays, the reaction was stopped with the addition of citric acid (1 μL) at a final concentration of 10 mM. All reagents were dispensed using the Multidrop™ Combi reagent dispenser (Thermo Fisher). Fluorescence was measured on a PHERAstar® (v5.41, BMG Labtech) using the FI 485 520 optic module. Data was normalised to 2% (v/v) DMSO (negative control, 0% inhibition) and 100% inhibition control (control compound was used where available, buffer excluding the enzyme was used if none were accessible).

Assay conditions were optimised to account for potent inhibitors **WEHI-P4** and **WEHI-P8**. Here, PLpro enzyme (final concentration 5 nM) was added to the plates and incubated at room temperature for 10 min. UbRh (final concentration 100 nM, except for ATXN3 where 2000 nM was used) was added to start the reaction and incubated for 120 min before stopping the reaction. The assay conditions for each enzyme are as follows and in this format (enzyme: final concentration (nM) / reaction time (min)): SARS-CoV-2 PLpro: 50 or 5 nM, 12 or 120 min; SARS-CoV-2 PLpro^BL: 50 nM, 12 min; SARS-CoV-PLpro: 20 nM, 12 min; MERS-CoV PLpro: 10 nM, 12 min; HKU1 PLpro: 0.2 nM, 3 min; OC43 PLpro: 2.5 nM, 3 min; NL63 PL2pro: 0.025 nM, 10 min; 229E PL2pro: 0.1 nM, 30 min; USP21: 5 nM, 2 min; ATXN3: 50 nM, 2 min; Cezanne: 0.5 nM, 2 min; OTUD1: 10 nM, 2 min; USP10: 50 nM, 12 min; UCHL3: 0.005 nM, 2 min.

**WEHI-P3 Specificity Assay (Ubiquigent).** **WEHI-P3** was assayed using the commercial UbRh-based DUBprofiler™ drug discovery screening platform and results were analysed and provided by Ubiquigent (Dundee, Scotland). SARS-CoV-2 PLpro protein and compound **WEHI-P3** were supplied to Ubiquigent.

**Surface plasmon resonance.** Experiments were performed on a BIAcore 8 K+ instrument (Cytiva, USA) PLpro proteins were diluted into HBS-P+ (see below) prior to immobilisation on a Sensor Chip SA (Cytiva, USA) by coupling. Compounds were spotted onto a Greiner 96-well U bottom plate (Item no. 650001) using the ECHO acoustic liquid dispenser from a 10 mM stock to desired concentrations and backfilled with DMSO to give a final DMSO of 2% (v/v). Compounds were further diluted into a dilution buffer consisting of 20 mM HEPES pH 7.4, 150 mM sodium chloride, 0.05% (v/v) P20 detergent (HBS-P+) and 1 mM TCEP. Running buffer consisted of dilution buffer supplemented further with 2% (v/v) DMSO (HBS-P+, 2% (v/v) DMSO). Multi-cycle kinetics were performed with 270 sec associations and 1800 s dissociations with no further regeneration. Binding constants were determined in BIAcore insight evaluation (version 3.0.12) at equilibrium averaging response over 5 sec with a midpoint 5 sec before the end of the association phase. Final $K_D$ values were determined by averaging the values from two independent experiments.

**SEC-MALS.** Size-exclusion chromatography multi-angle light scattering (SEC–MALS) experiments were performed using a Superdex 200 Increase 10/300 GL column (Cytiva) coupled with DAWN HELEOS II light scattering detector and Optilab T-rEX refractive index detector (Wyatt Technology). The system was equilibrated in 50 mM Tris

(pH 7.5), 50 mM NaCl, 2% (v/v) DMSO running buffer and calibrated using bovine serum albumin (2 mg/mL) before analysis of experimental samples. The system was then equilibrated again with running buffer where DMSO was replaced with 100 μM of the respective compound before each run. For each experiment, 50 μL of purified protein (2 mg/mL) was injected onto the column and eluted at a flow rate of 0.5 mL/min. Experimental data were collected and processed using ASTRA (Wyatt Technology, v.7.3.19)

**FRET assay.** A cellular assay to test compound efficacy was used as described[30]. For details on tissue culture and cell line verification see **Part 3** below.

Briefly, we constructed a HEK293T cell line stably expressing a FRET biosensor composed of mClover3 donor and mRuby3 acceptor fluorophores separated by a linker containing PLpro cleavage motif $T_{P-5}L_{P-4}K_{P-3}G_{P-2}G_{P-1} \downarrow A_{P-1}P_{P-2}T_{P-3}K_{P-4}V_{P-5}$. The cell line was then lentivirally transduced with PLpro coding sequencing embedded in a Tet-On expression vector, allowing PLpro expression to be controlled by the addition of dox. This cell line, expressing both the FRET bionsensor, and PLpro under dox control, was used for drug screening.

We prepared 7 titrations of compounds in 3-fold dilution and seeded 1.5 μL of each onto wells of a 96-well flat-bottom plate. We also seeded 1.5 μL DMSO as a no-treatment control. Next, $2.5 \times 10^5$ cells in 150 μL media with 300 ng/mL dox were added into those wells and incubated at 37 °C, 10% $CO_2$ overnight. Additional wells were prepared with 1.5 μL DMSO and $2.5 \times 10^5$ cells in 150 μL media without dox as no dox control. Cells were detached and analysed by flow cytometry (WEHI FACS facility) to determine the FRET+ percentage.

We fit the dose-response curves in prism10 using Eq. 1 (below):

$$FRET\% = Bottom + \frac{Top - Bottom}{\frac{EC50}{[\text{inhibitor concentration}]} + 1} \qquad (1)$$

FRET% from no treatment was assigned a concentration of 1 nM to capture the baseline without treatment. If complete inhibition was not observed from the top two concentrations, FRET% from no-dox treatment was assigned with a concentration of 0.1 mM and included for curve fitting. The reported $EC_{50}$ was used to evaluate compound efficiency.

**Crystallography.** Crystallisation screening was performed at the CSIRO's Collaborative Crystallisation Centre (C3) or the Monash Macromolecular Crystallisation Platform in Melbourne, Australia. SARS-CoV-2 PLpro complex crystals were generated by incubation of SARS-CoV-2 PLpro[C111S] (13 mg/mL) with 0.44 mM inhibitor (0.88% (v/v) DMSO final), overnight at 4 °C and precipitate removed by centrifugation prior to dispensing.

**Crystallisation of SARS-CoV-2 PLpro[C111S]-WEHI-P1.** Crystals grew from a reservoir containing 0.2 M sodium succinate, 0.1 M trisodium citrate-citric acid pH 5.72, 10% PEG 8 K (w/v) at 8 °C in a sitting-drop vapour-diffusion experiment (150 nL protein to 150 nL reservoir solution). Crystals were cryoprotected with reservoir solution supplemented with 17% (v/v) PEG400 and 0.44 mM inhibitor prior to flash freezing in liquid nitrogen.

**Crystallisation of SARS-CoV-2 PLpro[C111S]-WEHI-P2.** Crystals grew from a reservoir containing 0.2 M sodium acetate, 0.1 M trisodium citrate-citric acid pH 5.4, 10% (w/v) PEG 8 K at 8 °C in a sitting-drop vapour-diffusion experiment (150 nL protein to 150 nL reservoir solution). Crystals were cryoprotected with reservoir solution supplemented with 17% (v/v) glycerol and 0.44 mM inhibitor prior to flash freezing in liquid nitrogen.

**Crystallisation of SARS-CoV-2 PLpro[C111S]-WEHI-P4.** Crystals grew from a reservoir containing 0.3 M sodium malonate, 0.1 M tris pH 7.6, 6% (w/v) PGA-LM (poly-γ-glutamic acid low molecular weight polymer), 50 μM $ZnCl_2$ at 4 °C in a sitting-drop vapour-diffusion experiment (150 nL protein to 150 nL reservoir solution). Crystals were cryoprotected with reservoir solution supplemented with 20% (v/v) glycerol and 0.44 mM inhibitor prior to flash freezing in liquid nitrogen.

**Crystallisation of SARS-CoV-2 PLpro[C111S]-WEHI-P24.** Crystals grew from a reservoir containing 0.2 M lithium acetate, 0.1 M trisodium citrate-citric acid pH 5.97, 10% (w/v) PEG 8 K at 8 °K in a sitting-drop vapour-diffusion experiment (150 nL protein to 150 nL reservoir solution). Crystals were cryoprotected with reservoir solution supplemented with 17% (v/v) glycerol and 0.44 mM inhibitor prior to flash freezing in liquid nitrogen.

**Data collection, phasing and refinement.** Diffraction data were collected at the Australian Synchrotron (Australian Nuclear Science and Technology Organisation, ANSTO) beamline MX2[61] (wavelength: 0.9537 Å, temperature: 100 K). Datasets were either auto-processed at the synchrotron using XDS[62], Aimless and Pointless[63,64] or using similar methods in the ccp4i2 'Data reduction task'[65]. Datasets were solved by molecular replacement in Phaser[66]. In the case of **WEHI-P4** and **WEHI-P1**, the apo structure of SARS-CoV-2 PLpro was used as a search model (PDB: 6WRH)[34]. For **WEHI-P2** and **WEHI-P24**, **WEHI-P1** with the ligand removed was used as a search model and its FreeR flags were used in the working dataset for refinement. Two rounds of simulated annealing were also conducted prior to model refinement to minimise model bias. Refinement and model building was performed in Phenix[67] and Coot[68]. TLS parameters were set to one TLS group per chain where appropriate. Additional NCS refinement was utilised in each refinement cycle. Geometric restrains for compounds were generated by the GRADE web server (http://grade.globalphasing.org) or using Phenix eLBOW[69]. Models were validated using MolProbity[70]. Final Ramachandran statistics for; **WEHI-P1** were 0.00% outliers, 2.21% allowed and 97.79% favoured; **WEHI-P2** were 0.00% outliers, 1.37% allowed and 98.63% favoured; **WEHI-P4** were 0.00% outliers, 2.93% allowed and 97.07% and **WEHI-P24** were 0.00% outliers, 2.04% allowed and 97.96% favoured. Structural figures were generated using ChimeraX[71]. Data collection and refinement statistics can be found in Supplementary Table 1.

**AlphaFold2.** Alphafold2[37] was used to generate a model for OC43-CoV PLpro[WT] (aa 1561-1872 of polyprotein 1ab, RefSeq: AY391777), 229E-CoV PL2pro[WT] (aa 1599-1905 of polyprotein 1ab, RefSeq: NP_073549) and NL63-CoV PL2pro[WT] (aa 1578-1876 of polyprotein 1ab, RefSeq: YP_003766). ColabFold[72] (ver1.5.5) was used to output five predicted models (relaxed) of which the number one ranked model for each protein was used in this manuscript. Source code was downloaded and run on internal servers.

**Molecular modelling and molecular dynamics simulations.** Modelling was performed using the Schrödinger suite (Release 2024-2: Maestro, Schrödinger, LLC, New York, NY, 2024). The crystal structure of **WEHI-P4** was used as the starting model. Prior to simulations all solvent molecules were removed and S111 mutated to the WT cysteine before the structure was prepared for simulations using the Protein Preparation Wizard[73]. The structure was prepared for molecular dynamics using the System Builder wizard with a TIP3P water model and an orthorhombic water box buffered to 15 Å in all directions. Sodium and Chloride Ions were placed to neutralise the model and ions added to a concentration of 0.15 M. Molecular dynamics simulations were performed using an OPLS4[74] forcefield in Desmond[75]. Initial simulations were run for 1.2 ns for 250 frames at a constant temperature and pressure of 300 K and 1.01325 bar (NPT ensemble), using the

Relax model system before the simulation protocol. After the initial 1.2 ns simulation, 40 ns, 1000 frame simulations were performed on the relaxed model in triplicate. In the case of **WEHI-P70**, the **WEHI-P4** crystal structure pose was edited to include the pyrazole moiety in place of the cyclohexanol, then the structure minimised using Prime[76] prior to performing molecular dynamics simulations using the protocol previously described. For NL63 PL2pro an AlphaFold2 prediction[37] was used (see above). The PL2pro AF2 model was prepared by modifying the apo Zinc Finger domain to coordinate a $Zn^{2+}$ ion, and the blocking loop (residues 251-257) was removed. The **WEHI-P4** crystal structure was aligned on the PL2pro compound binding site and the blocking loop conformation from **WEHI-P4** merged into the PL2pro structure and mutated to NL63 sequence. The **WEHI-P4** pose was merged into the PL2pro model, and the model was minimised in Prime[76] prior to performing the simulation protocols described for PLpro above. Chosen frames from the 40 ns simulations represent a consensus conformation from the simulations.

**PLpro sequence alignments.** Annotated PLpro domains from Orf1ab (see **Molecular Biology** for accession codes) were extracted and a multiple sequence alignment (MSA) performed using CLUSTAL Omega (EMBL-EBI). The outputted sequence identity table and MSA appears in Supplementary Figs. 3 and 4 respectively. To generate the MSA Figure the alignment was inputted into ESPript 3.0 server[77] and annotated in Adobe Illustrator CC 2024. Sequences were visually assessed using the MSA and predicted structured (AF2) to determine appropriately aligned residues.

## Medicinal chemistry and DMPK studies

Details on medicinal chemistry including synthesis and compound characterisation used in this study can be found in the Supplementary Information.

**Kinetic solubility.** Kinetic solubility of compounds was determined based on a method described previously[78]. Test compounds prepared at 10 mg/mL in DMSO were diluted into buffer (pH 2.0 or pH 6.5) to give a 1% (v/v) final DMSO concentration. After standing for 30 min at ambient temperature, samples were analysed via nephelometry to determine a solubility range. The maximum value of the assay is 100 μg/mL and the minimum value is 1.6 μg/mL.

**Partition co-efficient estimation and physicochemical properties.** Partition coefficient values (LogD) were estimated at pH 7.4 by correlation of their chromatographic retention properties against the characteristics of a series of standard compounds with known partition coefficient values. The method employed gradient HPLC based on a previously published method[79]. Physicochemical properties for drug-likeness calculated using the ChemAxon for Excel software (ver. 20.21.0.768).

**Microsome Stability.** Mouse (lot #2210246) and human liver microsomes (lot # 1910096) were sourced from XenoTech LLC, Kansas City, KS. The microsomal stability assay was performed by incubating compounds (0.5 μM) with human or mouse liver microsomes (0.5 mg/mL), suspended in 0.1 M potassium phosphate buffer (pH 7.4) containing 3.3 mg/mL $MgCl_2$ at 37 °C. The metabolic reaction was initiated by the addition of NADPH (to give 1.3 mM). Control samples in the absence of cofactor were also included. Samples were mixed and maintained at 37 °C using a microplate incubator (THERMOstar ®, BMG Labtech GmbH, Offenburg, Germany) and quenched at various time points over 60 min by the addition of MeCN containing an internal standard. Quenched samples were centrifuged, and the supernatant removed and analysed by LC/MS (Waters Xevo G2 QToF MS coupled to an Acquity UPLC) using a Supelco Ascentis Express RP C8 column (5 cm × 2.1 mm, 2.7 μm) and a mobile phase consisting of 0.05% (v/v) FA in $H_2O$ and 0.05% (v/v) FA in MeCN and mixed under gradient conditions. The flow rate was 0.4 mL/min and injection volume was 5 μL. The in vitro intrinsic clearance was calculated from the first-order degradation rate constant for substrate depletion.

**Hepatocyte Stability.** Mouse (lot # 2310051) and human cryopreserved hepatocytes (lot # 2310092) were sourced from XenoTech LLC, Kansas City, KS. The hepatocyte stability assay was performed on a plate shaker (900 rpm) placed in a humidified incubator set at 37 °C with 7.5% $CO_2$ atmosphere and ~95% RH. Cryopreserved hepatocytes were suspended in protein-free Krebs-Henseleit buffer (KHB; pH 7.4) at a concentration of 0.5 million viable cells/mL. The hepatocyte cell viability was assessed using Trypan blue dye exclusion method. The metabolic reaction was initiated by addition of compounds (test or QC cocktail) to aliquots of hepatocyte suspension that were pre-equilibrated (for 10 min) at 37 °C and 7.5% $CO_2$. At various time points over 240 min, samples were quenched by addition of MeCN containing an internal standard. Quenched samples were left on ice for approximately 15 min, centrifuged and the supernatant removed and analysed by tandem quadrupole-Time of Flight MS (Waters G2 QToF) with a mass range scan of 50-1200 Da. The in vitro intrinsic clearance (μL/min/$10^6$ cells) was calculated from the first order degradation rate constant.

**Protein plasma binding.** Mouse (CD1, pooled, mixed gender, Na Heparin as anticoagulant; lot # MSE433327) and human plasma (pooled, mixed gender, Na Heparin as anticoagulant; lot # HMN921520) was sourced from BioIVT, Hicksville, NY. Plasma protein binding was conducted by rapid equilibrium dialysis (RED) using a modification of a method published previously[80]. Briefly, plasma was spiked with compound, mixed, and aliquots taken to plasma. The remaining spiked plasma was equilibrated at 37 °C (~10 min) prior to adding to the RED inserts (300 μL per insert). Inserts (n = 4) were placed in a teflon holding plate and dialysed against protein-free buffer (500 μL per insert) at 37 °C on an orbital plate shaker (ThermoMixer C, Eppendorf; 800 rpm). To control the pH of the assay matrix, the dialysis was performed in an incubator under a humidified $CO_2$-enriched atmosphere; the pH of post-dialysis plasma and dialysate was confirmed to be within pH 7.4 ± 0.1. At the end of the dialysis period, aliquots were taken from the donor and dialysate chambers to obtain measures of the total and free compound concentration, respectively. To allow quantification using a single calibration curve, each sample was mixed with an equivalent volume of the opposite medium (i.e. blank assay matrix for dialysate samples and blank dialysate medium for donor samples). The matrix-matched samples were stored at -80 °C until analysis by LC-MS. For the stability assessment, residual spiked plasma was incubated at 37 °C in parallel to the RED samples. Aliquots were taken at 3 and 6 h, mixed with an equivalent volume of blank dialysate medium, snap frozen on dry ice and at -80 °C until analysis by LC-MS.

Quantitation was performed following protein precipitation with MeCN (2 to 1 volume ratio relative to the matched matrix samples) and separation of the supernatant. Samples were analysed by mass spectrometry using a SCIEX Triple Quad 6500+ mass spectrometer coupled to an Echo module for sample ejection. Detection was by positive electrospray ionisation with multiple reaction monitoring. The carrier solvent was 0.1% (v/v) FA, 1 mM ammonium fluoride, 0.5 mM citric acid in 70% (v/v) MeCN/$H_2O$ with a 400 μL/min flow rate. Quantitation was by comparison of the response to calibration standards prepared in the same matrix and processed using the same method. Assay acceptance was based on the calibration range (2.5-2000 ng/mL) and accuracy and precision at low, mid and high concentrations.

**Mouse exposure after oral dosing of 100 mg/kg.** The systemic exposure of **WEHI-P8** was studied in non-fasted male C57BL/6 mice

weighing 20.5 – 22.6 g. Mice had access to food and water ad libitum throughout the pre- and post-dose sampling period. On the day of dosing, solid compound was dispersed in pre-mixed 0.5% (w/v) methylcellulose (Methocel A4M) in Milli-Q water using vortexing and sonication, creating a uniform fine white suspension with an apparent pH of 7.2. The bulk formulation was mixed by inverting the tube prior to drawing each dosing volume. Dosing was by oral gavage (10 mL/kg) and blood samples were collected up to 24 h ($n = 2$-3 mice per time point) with a maximum of three samples from each mouse via sub-mandibular bleed (approximately 120 µL; conscious sampling). Blood was collected into polypropylene Eppendorf tubes containing heparin as anticoagulant, centrifuged immediately, supernatant plasma was removed, and stored at -80 °C until analysis by LC-MS. Just prior to analysis, proteins were precipitated using MeCN at a 1:3 volume ratio (plasma to MeCN) and samples were centrifuged and the supernatant injected onto the LC/MS system.

Processed samples were analysed using a Waters Xevo TQD coupled to a Waters Acquity UPLC with positive electrospray ionisation and multiple reaction monitoring. The column was a Supelco Ascentis Express RP C8 column (50 × 2.1 mm, 2.7 µm) maintained at 40 °C and the mobile phase was 0.05% (v/v) FA in $H_2O$ and 0.05% (v/v) FA in MeCN mixed by gradient elution from 15 to 75% (v/v) MeCN over 2 min with a flow rate of 0.8 mL/min. Quantitation was by comparison to calibration standards prepared in blank mouse plasma and processed as for the samples. Diazepam was included as an internal standard in both samples and calibration standards. The assay was validated for calibration range (1-5000 ng/mL), lower limit of quantitation (1 ng/mL), accuracy, (within ± 10%) and precision (relative standard deviation of <10%). Plasma concentration versus time data were analysed using non-compartmental methods.

**CYP inhibition.** The CYP inhibition assay was performed with human liver microsomes utilising a substrate-specific interaction approach which relies on the formation of a metabolite that is mediated by a specific CYP isoform. The assay conditions employed for each CYP isoform are based on that previously reported[81]. Phosphate buffer (0.1 M) was prepared by dissolving monobasic potassium phosphate ($KH_2PO_4$) and dibasic potassium phosphate ($K_2HPO_4$) in 500 mL deionised water and adjusting pH to 7.4. Magnesium chloride was added at 3.3 mM to prepare the final incubation buffer.

A suspension of human liver microsomes was prepared in the incubation buffer at the required protein concentration. Multiple concentrations of test compound and positive control inhibitors were incubated with human liver microsomes at 37 °C concomitantly with each substrate. The total organic solvent concentration was kept at 0.5% (v/v). The reactions were initiated by the addition of NADPH (final concentration 1.3 mM) and the samples were quenched by the addition of ice-cold acetonitrile containing internal standard (0.15 µg/mL of diazepam). Concentrations of the substrate-specific metabolites in quenched samples were determined by UPLC-MS relative to calibration standards prepared in quenched microsomal matrix. Control samples were included to assess whether the UPLC-MS assay of the specific metabolites was affected in the presence of each test compound (and potential metabolites). Positive control compounds for each CYP are outlined in the Supplementary Information.

**Time dependent inhibition.** The time-dependent CYP inhibition assay was performed with human liver microsomes utilising a substrate-specific interaction approach which relies on the formation of a metabolite that is mediated by CYP3A4/5. The assay conditions employed are based on that previously reported[82].

**hERG study.** hERG binding assessment was carried out with *WEHI-P8* in 10-pt dose $IC_{50}$ mode at Reaction Biology (Malvern, PA) using a Predictor™ hERG Fluorescence Polarisation Assay[83].

## Studies in cells and in vivo

Reagents include antibodies against CD3 (1:500, Agilent A045201), MPO (1:1000, Agilent A039829), F4/80 (1:1000, WEHI in-house antibody) or SARS-CoV-2 nucleocapsid (1:4000, abcam ab271180) using the automated Omnis EnVision G2 template (Dako, Glostrup, Denmark). Chemical reagents include the Pgp inhibitor CP-100356 (Sigma Aldrich PZ0171), 4% paraformaldehyde (PFA) in PBS (Thermo J61899-AK) and 2-chloroacetamide (Sigma Aldrich C0267).

**Tissue culture and cell line verification.** HEK293T cells (FRET assay) were authenticated and sourced from CellBank Australia.

VERO cells were purchased from ATCC (clone CCL-81). Calu-3 and Vero (CCL-81) cells displayed expected cell morphologies and were sent for validation to Garvan Molecular Genetics facility (on 15 June 2020). Cell lines were screened on a monthly basis for *mycoplasma* contamination using the PlasmoTest kit (Invitrogen) as per the manufacturer's instructions. All used cells were *mycoplasma* free.

**Measurement of Calu-3 in vitro 50% tissue culture infectious dose (TCID50).** For infection assays Calu-3 cells were seeded in a volume of 100 µL DMEM F12 into tissue culture-treated flat-bottom 96-well plates (Falcon) at a density of $3.5 \times 10^4$ cells/well and incubated over night before infection and/or treatment at confluency. On day of infection and/or treatment cells were washed twice with serum-free DMEM medium and infected with SARS-CoV-2 clinical isolate VIC001[84] and MOI of 0.1 in 25 µL of serum-free medium containing TPCK trypsin (0.5 µg/mL working concentration, ThermoFisher). Cells were cultured at 37 °C and 5% $CO_2$ for 30 min. Cells were topped up with 150 µL of medium containing PLpro inhibitor compounds at indicated concentrations in 6 replicates per concentration. At 48 h post infection/treatment, 100 µL of supernatant was harvested from each well and kept frozen at -80 °C. For TCID50 assays, Vero cells were seeded in a volume of 100 µL DMEM medium into tissue culture-treated flat-bottom 96-well plates (Falcon) at a density of $1 \times 10^4$ cells/well and incubated overnight. The next day, Vero plates were washed twice with PBS and 125 µL of DMEM + 100 U/mL penicillin and 100 mg/mL streptomycin (serum free) + TPCK trypsin (0.5 µg/mL working conc) was added and kept at 37 °C, 5% $CO_2$. Calu-3 cell supernatants were thawed and serial 1:7 dilutions were prepared in 96-well round bottom plates at 6 replicates per dilution. 25 µL of serially diluted calu-3 supernatant were added onto Vero cells and plates incubated for 4 days at 37 °C, 5% $CO_2$ before measuring cytopathic effect under a light microscope. The TCID50 calculation was performed using the Spearman and Kärber method[85].

**Plaque assay.** Plaque assay was adapted and performed based on protocols previously described[86]. Briefly, African green monkey kidney epithelial Vero cells, purchased from ATCC (clone CCL-81), were seeded in flat bottom 24-well plates ($8 \times 10^4$ cells/well) and left to adhere overnight at 37°C/5% $CO_2$. Cells were washed twice with PBS and transferred to serum-free DMEM containing TPCK trypsin (0.5 µg/mL working concentration). Cells were infected with 150 µL of SARS-CoV-2 clinical isolate VIC001 (TCID50 $2.6 \times 10^3$/mL) and incubated at 37°C/5% for 30 min. Next, 150 µL of 1:2 serial dilutions of the hit compounds ranging from final concentrations of 5 µM to 0.0098 µM with or without 2 µM of the P-glycoprotein inhibitor CP100356 were transferred to the infected cells and incubated at 37°C/5% $CO_2$ for 30 min. Cells were then overlayed with 1.5% (w/v) methylcellulose and 4% FCS (v/v) in DMEM and incubated at 37°C/5% $CO_2$ for 4 days. At 4 dpi the overlay was removed, and cells were washed once with PBS before fixation with 4% paraformaldehyde (PFA) (v/v) in PBS for 40 min at room temperature. Wells were then stained with 0.2% crystal violet (w/v) in 20% methanol (v/v) for 10 min, then washed twice with MilliQ water and air dried before plaque counting and calculation of antiviral $EC_{50}$ for each compound using four-parameter logistic regression using GraphPad Prism 8.0 (GraphPad Software Inc).

**Ethics statement.** In vivo efficacy and long COVID studies were performed at The Walter and Eliza Hall Institute of Medical Research (WEHI). Procedures and mouse strains were reviewed and approved by the WEHI Animal Ethics Committee (ethics number 2020.016 and 2024.006). Mouse exposure studies were conducted at Monash Institute using established procedures in accordance with the Australian Code of Practice for the Care and Use of Animals for Scientific Purposes, and were reviewed and approved by the Monash Institute of Pharmaceutical Sciences Animal Ethics Committee (ethics protocol number 26789). All animal experiments were conducted in accordance with the Prevention of Cruelty to Animals Act (1986) and the Australian National Health and Medical Research Council Code of Practice for the Care and Use of Animals for Scientific Purposes (1997).

**Mice.** Male or female WT C57BL/6 J mice were bred and maintained in the Specific Pathogen Free (SPF) Physical Containment Level 2 (PC2) Bioresources Facility at The Walter and Eliza Hall Institute of Medical Research (WEHI).

All procedures involving animals and live SARS-CoV-2 strains were conducted in an OGTR-approved Physical Containment Level 3 (PC3) facility at WEHI (Cert-3621). Mice were transferred to the PC3 laboratory for all SARS-CoV-2 infection experiments at least 4 days prior to the start of experiments. Animals were age- and sex-matched within experiments (both sexes were used). Experimental mice were housed in individually ventilated microisolator cages under level 3 biological containment conditions with a 12-h light/dark cycle and provided standard rodent chow and sterile acidified water *ad libitum*.

**SARS-CoV-2 strains.** SARS-CoV-2 VIC2089 clinical isolate (hCoV-19/Australia/VIC2089/2020) was obtained from the Victorian Infectious Disease Reference Laboratory (VIDRL). Viral passages were achieved by serial passage of VIC2089 through successive cohorts of young C57BL/6 J (WT) mice[38]. Briefly, mice were infected with SARS-CoV-2 clinical isolate intranasally. At 3 dpi, mice were euthanised and lungs were harvested and homogenised in a Bullet Blender (Next Advance Inc) in 1 mL Dulbecco's modified Eagle's medium (DMEM) (Gibco/ThermoFisher) containing steel homogenisation beads (Next Advance Inc). Samples were clarified by centrifugation at 10,000 x g for 5 min before intranasal delivery of 30 μL lung homogenate into a new cohort of naïve C57BL/6 J mice. This process was repeated a further 20 times to obtain the SARS-CoV-2 VIC2089 P21 isolate. Lung homogenates from all passages were stored at -80°C.

**Infection of mice with SARS-CoV-2.** Mice were anesthetised with methoxyflurane and inoculated intranasally with 30 μL SARS-CoV-2. Virus stocks were diluted in serum-free DMEM to a final concentration of $10^4$ TCID50/mouse. After infection, animals were visually checked and weighed daily for a minimum of 10 days. Mice were euthanised at the indicated times post-infection by $CO_2$ asphyxiation. For histological analysis, animals were euthanised by cervical dislocation. Lungs were collected and stored at -80°C in serum-free DMEM until further processing.

SARS-CoV-2 P21 infections were performed with animals of 3 different aged groups depending on experimental outcome: young = 6-8 week-old; adult = 9-12 week-old; aged > 6 month-old. All animals were monitored and weighed daily, for a minimum of 10 days post-infection. Animals older than 10 weeks infected with SARS-CoV-2 P21 may require euthanasia due to excessive weight loss or extreme signs of disease. Humane points include weight loss greater than 20% of initial weight, and signs of lack of grooming, decreased body condition score, sustained weight loss (>15% over 3 consecutive days), laboured breathing, lethargy and/or decreased mobility. For PASC cohorts, where animal survival beyond the acute infection phase is essential for experimental outcomes, supportive care measures were implemented to minimize losses. These included saline administration upon reaching 15% weight loss and providing mashed food to prevent further deterioration. For the analysis of PASC phenotypes, animals were euthanized between 1 and 3 months post-infection (mpi). The 1 mpi cohorts correspond to 30–45 days post-infection (dpi), while the 3 mpi cohorts correspond to 75–90 dpi.

**In vivo antiviral treatment.** C57BL/6 (WT) mice were treated with either vehicle (10% DMSO in corn oil), Paxlovid-like treatment (56 mg/kg nirmatrelvir (MedChemExpress, HY-138687) + 19 mg/kg ritonavir (MedChemExpress, HY-90001)), 100 mg/kg or 150 mg/kg **WEHI-P8**. Acute infection experiments were performed either with a post-infection regime (6, 24 and 48 h post-infection) or starting 2 h pre-infection, followed by 6 and 24 h post-infection. PASC experiments were performed following the pre-infection regime.

**Measurement of lung viral loads via 50% tissue culture infectious dose (TCID50).** TCID50 was performed as previously described[85]. Briefly, African green monkey kidney epithelial Vero cells, purchased from ATCC (clone CCL-81), were seeded in flat bottom 96-well plates ($1.75 \times 10^4$ cells/well) and left to adhere overnight at 37°C/5% $CO_2$. Cells were washed twice with PBS and transferred to serum-free DMEM containing TPCK trypsin (0.5 μg/mL working concentration). Infected organs were defrosted, homogenised, clarified by centrifugation at 10,000 x g for 5 min at 4°C and supernatant was added to the first row of cells at a ratio of 1:7, followed by 9 rounds of 1:7 serial dilutions in the other rows. Cells were incubated at 37°C/5% $CO_2$ for 4 days until virus-induced cytopathic effect (CPE) was scored. TCID50 was calculated using the Spearman & Kärber algorithm[85].

**Histological analysis and immunohistochemical staining.** Organs were harvested and fixed in 4% paraformaldehyde (PFA) (v/v) for 24 h, followed by 70% ethanol (v/v) dehydration, paraffin embedding and sectioning. Slides were stained with either haematoxylin and eosin (H&E), or immunohistochemically with antibodies against CD3 (1:500, Agilent A045201), MPO (1:1000, Agilent A039829), F4/80 (1:1000, WEHI in-house antibody) or SARS-CoV-2 nucleocapsid (1:4000, abcam ab271180) using the automated Omnis EnVision G2 template (Dako, Glostrup, Denmark). Dewaxing was performed with Clearify Clearing Agent (Dako) and antigen retrieval with EnVision FLEX TRS, High pH (Dako) at 97 °C for 30 min. Primary antibodies were diluted in EnVision Flex Antibody Diluent (Dako) and incubated at 32 °C for 60 min. HRP-labelled secondary antibodies (Invitrogen, Waltham, USA) were applied at 32 °C for 30 min. Slides were counter-stained with Mayer Haematoxylin, dehydrated, cleared, and mounted with MM24 Mounting Medium (Surgipath-Leica, Buffalo Grove, IL, USA). Slides were scanned with an Aperio ScanScope AT slide scanner (Leica Microsystems, Wetzlar, Germany). An American board-certified pathologist (Smitha Rose Georgy) performed a qualitative analysis of H&E staining.

For Sirius Red staining, lung sections of 4 μm were dewaxed and rehydrated, followed by fixation in Bouin's fixative with heat for 1 h. Weigert's haematoxylin was used to stain nuclei before placing the sections into Picro-sirius red solution for 1 h, to stain for collagen. Stained sections were differentiated in acidified water, before dehydration, clearing and mounting with DPX. Images were acquired using an Olympus SlideView VS200 whole slide scanner, under 20x objective in brightfield mode.

**Scoring of H&E stained intestine sections.** Blinded histopathology scoring[87] of hematoxylin and eosin (H&E)-stained lungs and intestine sections was performed. For lungs, areas of haemorrhage and inflammation were scored by a researcher blinded to the experimental groups. Scoring ranged from 0, indicating no observable pathology, to 5, representing extensive pathology affecting the majority of the lung tissue. For intestines, scores were recorded for the proximal, middle

and distal colon, and proximal and distal small intestine. No epithelial damage was scored as 0, hyperproliferation of crypts was scored as 1, less than 50% crypt loss was scored as 2, more than 50% crypt loss was scored as 3, 100% crypt loss was scored as 4 and the presence of an ulcer was scored as 5. The presence of inflammatory cells in the mucosa, submucosa and muscle was scored separately. The presence of occasional inflammatory cells was scored as 0, increased mild numbers of inflammatory cells was scored as 1, moderate inflammatory cell presence was scored as 2 and severe inflammation was scored as 3. The scores of proximal and distal were summed to reveal the total histological score of small intestines, and for large intestines, the whole length was analysed, and the score is a sum of proximal, middle and distal sections. Average scores for an individual animal are presented +/- SEM.

**Lung cytokine and chemokine analysis.** Lungs were thawed, homogenised and clarified by centrifugation at 10,000 x $g$ for 5 min at 4°C. Supernatants were pre-treated for 20 min with 1% Triton-X-100 (v/v) for viral deactivation and the Cytokine & Chemokine 26-Plex Mouse ProcartaPlex Panel 1 (EPX260-26088-901) was used as described in the manufacturer's manual. 25 μL of clarified lung samples were diluted with 25 μL universal assay buffer, incubated with magnetic capture beads, washed, incubated with detection antibodies and SA-PE. Cytokines were recorded on a Luminex 200 Analyser (Luminex) and quantitated by comparison to a standard curve. Analysis was performed using R Studio.

**Proteomics.** Lungs of mock and infected animals were washed three times with ice-cold TBS, lysed in 2% sodium deoxycholate (SDC) (v/v) and 100 mM Tris-HCl (pH 8.5), and boiled immediately. After sonication, protein amounts were adjusted to 20 μg using the BCA protein assay kit. Samples were reduced with 10 mM (TCEP), alkylated with 40 mM 2-chloroacetamide, and digested with trypsin and lysC (1:50, enzyme/protein, w/w) overnight. Peptides were desalted using SDB-RPS-stage tips. Peptides were resolubilised in 5 μL 2% (v/v) acetonitrile (ACN) and 0.3% (v/v) trifluoroacetic acid (TFA) and 200 ng were injected into the mass spectrometer.

**LC-MS.** Samples were loaded onto a C18 fused silica column (inner diameter 75 μm, OD 360 μm × 15 cm length, 1.6 μm C18 beads) packed into an emitter tip (IonOpticks) using pressure-controlled loading with a maximum pressure of 1500 bar with the Neo Vanquish liquid chromatography system (Thermo Fisher Scientific) coupled to the MS (Orbitrap Astral, Thermo Fisher Scientific). Peptides were introduced onto the column with buffer A (0.1% FA) and 4% buffer B (80% ACN, 0.1% FA) followed by an increase of buffer B to 34% for 20 min, and 100% for 3 min at a flow rate of 400 nL/min.

A data-independent acquisition MS method was used in which one full scan (380–980 m/z, R = 240,000) at a target of $5 \times 10^6$ ions was first performed, followed by 300 windows with a resolution of 80,000 (at m/z 524) where precursor ions were fragmented with higher-energy collisional dissociation (collision energy 25%) and analysed with an AGC target of $8 \times 10^4$ ions and a maximum injection time of 3 ms in profile mode using positive polarity.

**Novel-object recognition test (NORT).** NORT[88] was performed to study object memory and preference for novelty. In short, mice were individually habituated to an empty testing arena (50 cm × 50 cm) for 10 min. On the next day, mice were placed in the same arena with two identical objects and allowed to freely roam for 10 min. A second trial was performed after an interval of 1 h in which mice were placed back into the testing area containing one of the familiar objects from trial 1 and one novel object. Mice were allowed to explore the testing arena for 5 min. The recognition index from trial 2 was calculated as a proportion of the time exploring the novel object over the total time spent exploring both objects.

**Quantification and statistical analysis**
Statistical analyses were performed using Prism v10.2.3 software (GraphPad Software, Inc.). Unpaired two-tailed t-tests were used for normally distributed data for comparisons between two independent groups. Data that violated the assumption of normality were transformed by generating $\log_{10}$ prior to statistical analysis. Bars in figures represent the mean or median (± SD or ±SEM) of normally or non-normally distributed datasets, respectively and as indicated in the Figure legends, and each symbol represents one mouse. Sample sizes (n), replicate numbers and significance can be found in the figures and figure legends.

Statistical analysis of cytokine data consisted of Wilcoxon rank sum test between group medians, with Bonferroni adjustment for multiple comparisons. Boxplots in figures depict the median and interquartile ranges. Loess smoothing was applied to the infection time course data, with the shaded area indicating 95% confidence intervals.

For proteome analysis, MS raw files were processed by the Spectronaut software version 19[89]. Mouse uniport FASTA databases (25,367 entries, 2021) were used as forward databases. Cysteine carbamidomethylation was included as fixed modification and N-terminal acetylation and methionine oxidations were included as variable modifications. The false discovery rate (FDR) and PEP cutoff were set to less than 1% at the peptide and protein levels and a minimum length of seven amino acids for peptides was specified. Enzyme specificity was set as C-terminal to arginine and lysine as expected using trypsin and LysC as proteases and a maximum of two missed cleavages. Statistical tests were performed with Perseus[90]. The 1D annotation-enrichment analysis detects whether expression values of proteins belonging to an enrichment term (here we used keywords, GOCC, GOMF, GOBP, and KEGG name) show a systematic enrichment or de-enrichment compared with the distribution of all expression values.

**Reporting summary**
Further information on research design is available in the Nature Portfolio Reporting Summary linked to this article.

# Data availability
All data is available within this paper or as Supplementary Information. **Source Data** is available for all Figures of the manuscript. Viral strains used in this study are available upon signing of a Materials Transfer Agreement. Genomic sequences of SARS-CoV-2 passages (P2 and 21) are available at GenBank, accession numbers OP848479-98 (https://www.ncbi.nlm.nih.gov/bioproject/PRJNA995787). The mass spectrometry proteomics data have been deposited to the ProteomeXchange Consortium via the PRIDE [1] partner repository with the dataset identifier PXD054356. Crystal structures have been submitted to the Protein Data Bank (PDB), accession numbers 9CYB **WEHI-P1**, 9CYC **WEHI-P2**, 9CYD **WEHI-P4**, 9CYK **WEHI-P24**. Source data are provided with this paper.

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

## Acknowledgements

The authors would like to thank present and past members of the Ubiquitin Signalling Division at WEHI. We would like to thank our CRO, Jubilant Biosys Ltd. (India) for their work in synthesising compounds

described in this work. The authors acknowledge the contribution and assistance of Melbourne Health through its Victorian Infectious Diseases Reference Laboratory at the Doherty Institute, as well as Prof Damian J Purcell and Dr Julie McAuley in providing isolated SARS-CoV-2 material including the clinical isolate that forms the basis of the presented animal model. We thank our supportive consumer team at WEHI. Importantly, we sincerely acknowledge the histology facility at WEHI for their exceptional support and expertise, which greatly contributed to the success of this study. This work has been supported by: a Wellcome Trust Innovator Award 222698/Z/21/Z to D.K., G.L., and M.J.C., WEHI Innovator funding to D.K., G.L., and M.J.C., MRFF MRF2002119 grants to D.K. and G.L., MRF2016781 to D.K., G.L., M.J.C., P.E.C., M.P., and S.A.C., MRF2032843 to A.J.H. and M.D; NHMRC Investigator Grants (GNT117812 to D.K., GNT2016461 to G.L., GNT1175011 to M.P. and GNT2009062 to P.E.C.); NHMRC Senior Research Fellowship (GNT1117089 to G.L.) and a donation to M.J.C. from John and Tibby Peterson. T.L.P. is supported by a Viertel Senior Medical Research Fellowship. K.S. and M.T. are supported by a Suzanne Cory Fellowship. We acknowledge Compounds Australia (www.compoundsaustralia.com) for their provision of specialised compound management and logistics research services to the project. The Australian Drug Discovery Library (ADDL) was funded by MTPConnect which is supported by the Australian Government, Department of Industry, Science and Resources. High throughput screening was performed at the National Drug Discovery Centre (NDDC), WEHI, Parkville, Australia, with support from the Australian Government Medical Research Future Fund (MRFF). The NDDC was supported by MRFF Grant EPCD000033, a donation from AWM Electrical, and a Victorian Government grant that funded laboratory equipment for Australian drug discovery research. Work in the laboratories of the authors was made possible through Victorian State Government Operational Infrastructure Support (OIS) and the Australian Government NHMRC Independent Research Institute Infrastructure Support (IRIIS) Scheme. This research was undertaken in part using the MX beamlines at the Australian Synchrotron, part of ANSTO, and made use of the Australian Cancer Research Foundation (ACRF) detector.

## Author contributions

D.K., G.L. and M.Doe conceived the project. D.K., G.L., M.Doe, S.M.D., S.A.C., M.J.C., J.P.M., K.N.L., P.E.C., M.P. and U.N. secured funding for this project. B.G.C.L., K.L., A.E.A., T.R.B., K.N.L. and J.P.M performed HTS screening. D.J.C., T.A.K., B.C.L. and D.K. performed in vitro studies and structural studies. R.W.B. and D.J.C. performed SPR and modelling studies. X.W. and M.J.C. performed cellular assays. S.M.B., R.B., L.M., M.D., J.S., L.S., A.T.S., J.P.C. C.C.A., G.E., K.C.D., M.P. and M.Doe undertook PC3, antiviral plaque and contributed to mouse studies. S.M.D., N.W.K, R.V., Y.K., J.P.M. and G.L. undertook medicinal chemistry studies. T.L.P. analyzed gut studies. E.A.K. and A.J.H were responsible for mice behavioural studies. G.C., K.K. and S.A.C. were responsible for DMPK studies. K.S. and M.T. performed and analyzed proteomics experiments. S.R.G. was responsible for histology. A.M. performed pathology. S.M.B., D.J.C., S.M.D., N.W.K., M.Doe and D.K. wrote the manuscript.

## Competing interests

This work is protected under provisional patent AU2024900559. The authors declare the following competing interests; D.K. is founder, shareholder and SAB member of Entact Bio and Proxima Bio. The remaining authors declare no competing interests.

## Additional information

Stefanie M. Bader ®[1,2,10], Dale J. Calleja ®[1,2,10], Shane M. Devine ®[1,2,10] ✉, Nathan W. Kuchel ®[1,10], Bernadine G. C. Lu ®[1,2], Xinyu Wu ®[1,2], Richard W. Birkinshaw ®[1,2], Reet Bhandari ®[1,2], Katie Loi[1,2], Rohan Volpe ®[1,2], Yelena Khakham[1], Amanda E. Au[1,2], Timothy R. Blackmore[1,2], Liana Mackiewicz ®[1], Merle Dayton[1], Jan Schaefer ®[1,2], Lena Scherer[1], Angus T. Stock[1,2], James P. Cooney[1,2], Kael Schoffer[1,2], Ana Maluenda[1], Elizabeth A. Kleeman ®[1,3], Kathryn C. Davidson[1,2], Cody C. Allison[1,2], Gregor Ebert ®[1,2], Gong Chen[4], Kasiram Katneni[4], Theresa A. Klemm[1,2], Ueli Nachbur[1,2], Smitha Rose Georgy[5], Peter E. Czabotar ®[1,2], Anthony J. Hannan ®[3,6], Tracy L. Putoczki[1,2,7], Maria Tanzer[1,2], Marc Pellegrini[1,2,9], Bernhard C. Lechtenberg ®[1,2], Susan A. Charman[4], Melissa J. Call ®[1,2], Jeffrey P. Mitchell ®[1,2], Kym N. Lowes[1,2], Guillaume Lessene ®[1,2,8] ✉, Marcel Doerflinger ®[1,2] ✉ & David Komander ®[1,2] ✉

[1]Walter and Eliza Hall Institute of Medical Research, Parkville, VIC, Australia. [2]Department of Medical Biology, University of Melbourne, Melbourne, Australia. [3]Florey Institute of Neuroscience and Mental Health, University of Melbourne, Parkville, VIC, Australia. [4]Centre for Drug Candidate Optimisation, Monash Institute of Pharmaceutical Sciences, Monash University, Parkville, VIC, Australia. [5]Anatomic Pathology – Veterinary Biosciences, Melbourne Veterinary

School, University of Melbourne, Werribee, VIC, Australia. [6]Department of Anatomy and Physiology, University of Melbourne, Parkville, VIC, Australia. [7]Department of Surgery, University of Melbourne, Melbourne, Australia. [8]Department of Biochemistry and Pharmacology, University of Melbourne, Melbourne, Australia. [9]Present address: Centenary Institute of Cancer Medicine and Cell Biology, Camperdown, NSW, Australia. [10]These authors contributed equally: Stefanie M. Bader, Dale J. Calleja, Shane M. Devine, Nathan W. Kuchel. ✉e-mail: devine.s@wehi.edu.au; glessene@wehi.edu.au; doerflinger.m@wehi.edu.au; dk@wehi.edu.au

