## [Transparent Peer Review file · Nature Communications]

A novel PLpro inhibitor improves outcomes in a pre-clinical model of long COVID

Corresponding Author: Professor David Komander

Version 0:

Reviewer comments:

Reviewer #1

(Remarks to the Author)

This paper presents a rigorous study focused on the development of antivirals targeting the PL protease of SARS-CoV-2. Importantly the lead compound shows promise in the mouse model by preventing long-covid like symptoms. The IC₅₀ values are 98 nM for SARS-CoV2 and an oral dose of 150mg/kg was used in the mice studies. This is a strong body of work with careful analysis. This work shows great promise towards the development of pan-CoV inhibitor, as they also study other alpha and beta CoVs in vitro.

I have previously reviewed this manuscript and in the current form, the authors have addressed all concerns expressed in my previous review.

Weaknesses:

One small issue was noted- Figure 3f, for the WEHI-8 treatment group, two scale bars are presented on each image.

Reviewer #2

(Remarks to the Author)

Bader et al have responded well to the comments of this reviewer. The manuscript describes a new inhibitor of PLpro of SARS-CoV-2 that also has efficacy against SARS-CoV, and to a lesser extent HCoV-NL63, and decreases the development of Long-COVID in mice.

One issue is that in their response to the reviewer comments (page 12, point #2, line 227), the authors state: "...its limited supply was prioritised for use for in vivo studies....". Can the authors comment on whether synthesizing large amounts of the final compound is feasible and economically practical?

Reviewer #3

(Remarks to the Author)

The revised submission by Bader et al. describes a SARS-CoV-2 PLP inhibitor. The revision is improved from the prior submission. There are still issues with the model and therapeutic administration that remain unclear. Since any SARS-CoV-2 direct acting antiviral with in vivo efficacy would similarly protect from "Long COVID" in these mouse models, the title of the paper is misleading because it implies that the molecule of interest is special in its ability to treat post-acute sequelae. Aside from that, the molecule is special in that it is a unique and potent inhibitor of SARS-CoV-2 PLP and this should be celebrated.

Incongruency with reported nirmatrelvir data? It is perplexing that 150mg/kg nirmatrelvir initiated prior to infection fails to exert an antiviral effect. Similarities and differences with published data should be discussed in this paper. The vehicle

utilized here (10% DMSO, corn oil) is a different flavour than the vehicle reported in Owen et al. Martinez et al (2024 STM, S10) shows a range of doses of nirmatrelvir (400, 120, 40 mpk) in therapy without ritonavir significantly reduces lung titer on 4dpi in BALB/c mice. The dose frequency of nirmatrelvir in prior preclinical studies noted above was BID (like the human dose regimen) yet here seems to be randomly infrequent. The rationale for non-uniform dose frequency is not clear. The differences with the data presented in the paper and the data in the literature should be discussed in the manuscript. The fairness of the comparison of PLT and WEHI-P8 is questionable given that PLT dose frequency was different than prior reports or that in humans. A rationale for this difference should be included in the paper.

Timing of therapy. The current draft assesses the impact of very early therapy (+6hr) on acute disease. Successful treatment of viral replication early not only abrogates acute disease severity but not surprisingly also impacts post-acute sequelae. In the rebuttal, the authors argue against performing additional more rigorous studies looking at initiating therapy at later more clinically relevant times and provide several examples of less rigorous reports in the literature with therapy which was also started at very early times. Please provide a rationale as to how +6hr therapy is a clinically relevant time to intervene in your model. As mentioned in 2023 Bader et al., viral replication peaks between day 2 and 4 with P21. Thus, you have time after 6hr within which to intervene prior to peak replication.

Clarity of your mouse adapted virus. Your P21 virus is an engineered, mouse adapted virus, which has been modified to enhance infectivity. In your case, the virus was the engineer. Through passage, your "human variant" has evolved and acquired mutations that increase replicative capacity and virulence in mice. Thus, the following statement should be edited for accuracy: "The recently reported mouse models of PASC rely on a genetically engineered viral strain, modified to enhance infectivity in mice and are not derived from human isolates. In contrast, our acute model of severe disease, is derived from an Australian circulating variant." Does it really matter how these viruses became mouse adapted or what the starting material was going into passage?

Reviewer #4

(Remarks to the Author)

The author have already revised the manuscript carefully according to the comments of reviewers. The questions of reviewers have been answered point to point. So, this revised version could be accepted to be published in nature communications.

Version 1:

Reviewer comments:

Reviewer #3

(Remarks to the Author)

"We don't believe there are any published studies demonstrating that a SARS-CoV-2 antiviral has efficacy against long COVID in animal models, let alone one that acts on PLpro." Don't Stop Believin'. See Figure 7:
<https://pmc.ncbi.nlm.nih.gov/articles/PMC9273046/>

Point-by-point response to reviewer comments

Nature submission #2024-07-14657.

Transferred to Nature Communications #NCOMMS-24-80357-T.

Original submission:

Bader, Calleja, Devine & Kuchel et al. "A novel PLpro inhibitor improves outcomes in a pre-clinical model of long COVID".

We would like to thank the Editor and the Reviewers for their continued constructive comments on our manuscript. We believe we have addressed the main points raised by the reviewers in our revised manuscript.

We have highlighted significant changes in the manuscript in **yellow** and provide the following point-by-point response to address the comments from the reviewers. Below we paraphrase reviewer and editorial comments, ***in italics and bold***, and our responses are in plain text.

We acknowledge the concerns of the editors and reviewer #3 about the brevity of our title and propose the following as a potential substitute:

We have also followed the advice of the editor and reviewer #3 and tempered the related conclusion around the direct relevance of our findings in a clinical setting. We have amended the Abstract as well as the Discussion accordingly to include limitations of our work that would be addressed in follow up studies.

REVIEWER COMMENTS

Reviewer #1 (Remarks to the Author):

This paper presents a rigorous study focused on the development of antivirals targeting the the PL protease of SARS-CoV-2. Importantly the lead compound shows promise in the mouse model by preventing long-covid like symptoms. The IC50 values are 98 nM for SARS-CoV2 and an oral dose of 150mg/kg was used in the mice studies. This is a strong body of work with careful analysis. This work shows great promise towards the development of pan-CoV inhibitor, as they also study other alpha and beta CoVs in vitro. I have previously reviewed this manuscript and in the current form, the authors have addressed all concerns expressed in my previous review.

Weaknesses:

One small issue was noted- Figure 3f, for the WEHI-8 treatment group, two scale bars are presented on each image.

We thank reviewer #1 for this comment and for the validation of our research. The error has been corrected.

Reviewer #2 (Remarks to the Author):

Bader et al have responded well to the comments of this reviewer. The manuscript describes a new inhibitor of PLpro of SARS-CoV-2 that also has efficacy against SARS-CoV, and to a lesser extent HCoV-NL63, and decreases the development of Long-COVID in mice.

One issue is that in their response to the reviewer comments (page 12, point #2, line 227), the authors state: "...its limited supply was prioritised for use for in vivo studies....". Can the authors comment on whether synthesizing large amounts of the final compound is feasible and economically practical?

At the time we had a limited supply of WEHI-P8 for all our studies, but this in no way is a reflection on the practicality and ease of which we produced this compound. The synthesis of WEHI-P8 and other WEHI-P analogues is relatively straight forward and mostly high yielding including with complex chiral purification. We have produced multigram quantities for this study of WEHI-P8 and our synthetic pathway is a feasible and economically practical means for even greater scale up in the future.

We thank reviewer #2 for their kind words and acknowledgment of our work as significant.

Reviewer #3 (Remarks to the Author):

The revised submission by Bader et al. describes a SARS-CoV-2 PLP inhibitor. The revision is improved from the prior submission. There are still issues with the model and therapeutic administration that remain unclear. Since any SARS-CoV-2 direct acting antiviral with in vivo efficacy would similarly protect from "Long COVID" in these mouse models, the title of the paper is misleading because it implies that the molecule of interest is special in its ability to treat post-acute sequelae. Aside from that, the molecule is special in that it is a unique and potent inhibitor of SARS-CoV-2 PLP and this should be celebrated.

We thank Reviewer #3 for their thoughtful comments. Reviewer #3 suggests that any SARS-CoV-2 direct-acting antiviral with sufficient in vivo efficacy would protect against long COVID. We don't believe there are any published studies demonstrating that a SARS-CoV-2 antiviral has efficacy against long COVID in animal models, let alone one that acts on PLpro. As demonstrated in our manuscript (Fig. 5), PLT was less effective than WEHI-P8 in mitigating post-acute lung and brain symptoms in mice in the dosing schedule we used which was based on experimental evidence specific to our in vivo model. Whether this difference stems from PLT's reduced in vivo efficacy in lowering viral burden in the P21 model compared to WEHI-P8 (Fig. 3d) or from the distinct roles of Mpro and PLpro remains an open question.

Nevertheless, we acknowledge that the original title did not fully capture the nuances of our findings and may have been misleading as we do not provide direct clinical evidence involving humans. To enhance clarity, we have revised the manuscript title to: "A novel PLpro inhibitor improves outcomes in a pre-clinical model of long COVID."

We have also expanded our discussion section to outline limitations and further directions to clarify the experimental outline and shortcomings of our study.

Incongruency with reported nirmatrelvir data? It is perplexing that 150mg/kg nirmatrelvir initiated prior to infection fails to exert an antiviral effect. Similarities and differences with published data should be discussed in this paper. The vehicle utilized here (10% DMSO, corn oil) is a different flavour than the vehicle reported in Owen et al. Martinez et al (2024 STM, S10) shows a range of doses of nirmatrelvir (400, 120, 40 mpk) in therapy without ritonavir significantly reduces lung titer on 4dpi in BALB/c mice. The dose frequency of nirmatrelvir in prior preclinical studies noted above was BID (like the human dose regimen) yet here seems to be randomly infrequent. The rationale for non-uniform dose frequency is not clear. The differences with the data presented in the paper and the data in the literature should be discussed in the manuscript. The fairness of the comparison of PLT and WEHI-P8 is questionable given that PLT dose frequency was different than prior reports or that in humans. A rationale for this difference should be included in the paper.

We thank the reviewer for their thorough interrogation of our treatment regimens. We respectfully disagree that it is perplexing that 150 mg/kg nirmatrelvir, when administered prophylactically, fails to exert a measurable antiviral effect. A comparison with the literature reveals key differences: In Owen et al., the daily dose required to achieve efficacy was four times higher than the dose we tested for nirmatrelvir alone. Additionally, we employed a different infection model as the one reported: in contrast to the genetically modified MA10 virus which was used in BALB/c mice in their study, our model relies on a naturally mouse-adapted virus containing the N501Y mutation which we used to infect C57BL/6 animals (see further explanations below). This distinction leads to differences in infectivity, as evidenced by our clinical isolate (P2), which results in 1–3 log higher viral burdens compared to MA-10 in the lungs. These differences in virus and host collectively contribute to the reduced efficacy of nirmatrelvir in our experimental system.

However, evaluating nirmatrelvir's efficacy in our mouse model was not the primary objective of our study, as this has been extensively characterized in the literature and is not used by itself to treat COVID-19 in humans. Instead, our focus was on reporting a novel PLpro inhibitor scaffold, and to compare our novel inhibitor to a combination of nirmatrelvir and ritonavir, which we believe better reflects the currently available and FDA-approved Paxlovid treatment. This represents a further key difference from published studies, which primarily assess the efficacy of nirmatrelvir alone. As highlighted in our previous rebuttal, the selected PLT dosing regimens were determined based on a series of experiments and calculations. The PLT mouse dose was scaled from the standard human dose of Paxlovid (300 mg nirmatrelvir, 100 mg ritonavir) administered to COVID-19 patients. This rationale is detailed in the manuscript (lines 272–282).

Although Owen et al. utilized a BID regimen of nirmatrelvir in their murine MA10 model, their dosing (300 and 1000 mg/kg) was significantly higher than the human-equivalent dose. In contrast, we

opted for a single PLT dose, as this was sufficient to reduce viral levels to the lower limit of detection (LOD) in our model, even when treatment was initiated 6 hours post-infection. This efficacy was comparable to WEHI-P8 at 150 mg/kg (see Figure here and Extended Data Fig 7d).

Given the distinct chemistry and molecular targets of the treatments being compared, we believe that determining dosing solely based on compound quantity would not constitute a fair comparison. Instead, we prioritized identifying the minimum effective dose required to reduce viral levels to the LOD in our experimental model.

In order to reflect the reviewers concern, we have now added, into the manuscript results section, our rationale for the treatment regimens, as well as added a limitation and further investigation section into the conclusion/discussion part.

Timing of therapy. The current draft assesses the impact of very early therapy (+6hr) on acute disease. Successful treatment of viral replication early not only abrogates acute disease severity but not surprisingly also impacts post-acute sequelae. In the rebuttal, the authors argue against performing additional more rigorous studies looking at initiating therapy at later more clinically relevant times and provide several examples of less rigorous reports in the literature with therapy which was also started at very early times. Please provide a rationale as to how +6hr therapy is a clinically relevant time to intervene in your model. As mentioned in 2023 Bader et al., viral replication peaks between day 2 and 4 with P21. Thus, you have time after 6hr within which to intervene prior to peak replication.

We appreciate the reviewer's concern that our study has limitations for direct clinical translation in regard to optimal dosing and timing, as has any pre-clinical animal study. Reviewer #3 refers multiple times to "clinically relevant" times, but in our opinion, this is in fact what we have attempted to address as there are key differences when comparing viral kinetics in our mouse model to infection to COVID-19 in humans. In our model, viral replication peaks at 1 dpi (see below) and is cleared by 7 dpi (see Bader et al., *PNAS* 2023 and also Bader et al., *CDDis* 2024). In contrast, viral titres in human SARS-CoV-2 infection peaks around day 7 and it can take as long as 40 days for virus to be cleared (Jang et al., *IJID* 2021). Please see below a direct comparison.

Infection kinetics in human COVID-19 patients (Jang et al., IJID 2021)

Infection kinetics of C57BL/6 mice with P21

While later treatment initiation might be clinically relevant for individuals who do not seek medical attention immediately, our study aims to evaluate the impact of an optimal, early intervention that is analogous to current human antiviral strategies. Collectively, we believe the fact that our inhibitor is highly efficacious in our mouse models of acute COVID - and when dosed as prophylactic, also in long COVID - is a very important step towards clinical translation. We acknowledge that further investigation towards human-relevant dosing, including later treatment timepoints, is required and now discuss this in the conclusion section. However, we believe including these experiments is beyond the scope of this current paper which focuses on discovery of a novel class of PLpro inhibitor and proof of principle for in vivo efficacy.

Clarity of your mouse adapted virus. Your P21 virus is an engineered, mouse adapted virus, which has been modified to enhance infectivity. In your case, the virus was the engineer. Through passage, your “human variant” has evolved and acquired mutations that increase replicative capacity and virulence in mice. Thus, the following statement should be edited for accuracy: “The recently reported mouse models of PASC rely on a genetically engineered viral strain, modified to enhance infectivity in mice and are not derived from human isolates. In contrast, our acute model of severe disease, is derived from an Australian circulating variant.” Does it really matter how these viruses became mouse adapted or what the starting material was going into passage?

We appreciate reviewer #3's thought experiment; however, in this context, we politely disagree that the virus itself is an “engineer.” In the case of our P21 virus, natural selection, not genetic engineering, drove adaptation as the virus did not deliberately alter its genetic code. Evolutionary pressure exerted by the host's immune system led to the selection of a viral subpopulation within P2 that was better suited to survive in the WT murine host. This process mirrors viral evolution in human populations, where SARS-CoV-2 naturally acquires mutations that enhance immune evasion. In contrast, genetically engineered viruses, such as MA10, incorporate targeted mutations that may not fully capture the complex evolutionary pathways a virus would take in vivo.

This artificially constrained adaptation can introduce biases, as engineered mutations are chosen based on prior assumptions rather than empirical fitness advantages.

Therefore, our statement is accurate. Genetic engineering refers to the deliberate modification of an organism's DNA using **laboratory-based technologies**, introducing mutations that do not occur naturally. By contrast, our approach relied on natural selection through serial passage, in which pre-existing viral variants within a diverse population were selected based on fitness in the murine host. Therefore, **P21 is not genetically engineered under any standard definition of the term.**

To address the reviewer's direct question: "Does it really matter how these viruses became mouse-adapted or what the starting material was?" **Yes, we believe that the method of adaptation significantly impacts the biological relevance of the model.** Natural selection through serial passage is superior to genetic engineering for generating physiologically relevant outcomes because it allows the virus to evolve under selective pressures that closely mimic real-world adaptation, rather than relying on predefined genetic modifications.

Reviewer #4 (Remarks to the Author):

The author have already revised the manuscript carefully according to the comments of reviewers. The questions of reviewers have been answered point to point. So, this revised version could be accepted to be published in nature communications.

We thank reviewer #4 for this positive endorsement of our work.